# Neural MRI : Differential Visualization of Hidden State Trajectories in Large Language Models

## Abstract

Neural MRI is a differential visualization framework for observing token-indexed hidden state trajectories in large language models. Given two aligned runs that differ only in a controlled input condition, we record token-indexed hidden states, compute per-position differences, and project the resulting sequence into a low-dimensional space using a deterministic linear projection (PCA). The projected sequence is rendered as an ordered trajectory by connecting successive token positions, yielding a comparable measurement object across experimental conditions. Using a fixed pipeline and consistent plotting conventions, we visualize trajectories for numeric token sequences and lexical input sets, showing coherent, ordered paths across diverse inputs. Our approach emphasizes reproducible measurement and comparison of internal state transitions while leaving functional interpretation to downstream analysis. This provides a model-agnostic method for observing hidden state transition structure in any instrumented LLM that exposes token-level activations.

## 1. Introduction

### 1.1. Background

Large language models (LLMs) have achieved strong performance across a wide range of tasks, yet their internal computations remain difficult to characterize. Most evaluation protocols treat an LLM as a black box, measuring quality primarily through input–output behavior (e.g., benchmark scores or human preference judgments). While this output-centric view is useful for tracking capability, it provides limited insight into how a model's internal state evolves as it processes a token sequence. As LLMs are increasingly deployed in settings where reliability and controllability matter, the ability to observe internal dynamics becomes an important scientific and engineering objective.

A central object of interest is the hidden state: a high-dimensional representation updated at each layer and token position during inference. Hidden-state analysis has motivated a broad literature, including probing classifiers, attention visualization, activation statistics, and representation similarity methods. These approaches have revealed correlations between internal activations and linguistic or task-related features, but they often focus on snapshots (points), aggregated distributions, or indirect readouts. In

particular, many methods do not explicitly isolate state transitions attributable to controlled input differences, and it remains challenging to visualize ordered trajectories that reflect token-by-token processing in a way that supports direct comparison across conditions.

This work targets the gap between performance evaluation and internal observation. Rather than inferring internal structure from external outputs, we seek a procedure that produces a measurable object directly from recorded hidden states, preserves token order, and can be applied consistently across different input types. Our goal is not to assign a unique semantic interpretation to internal coordinates, but to provide a reproducible way to visualize how internal representations change under controlled differences. Crucially, we treat these changes as a measurable object: a directly observable, token-ordered trajectory of condition-induced hidden-state transitions, rather than static points, abstracted structures, or aggregate summaries.

### 1.2. Research Question

We ask a single question: Can the transition structure of hidden states be made observable through an explicit operation and a standardized visualization procedure? Concretely, given two runs that differ only in a controlled input condition, can we extract and visualize an ordered sequence that reflects the induced change in hidden states across token positions, in a manner that is reproducible and comparable across diverse input types?

[1]Anonymous Institution, Anonymous City, Anonymous Region, Anonymous Country. Correspondence to: Anonymous Author <anon.email@domain.com>.

Preliminary work. Under review by the International Conference on Machine Learning (ICML). Do not distribute.

### 1.3. Contributions

We make three contributions:

1. **Differential hidden-state analysis.** We define a simple per-position differencing operation,

$$\Delta h(t) = h_A(t) - h_B(t),$$

computed from two aligned runs that differ only in a controlled input condition. This produces a sequence of displacement vectors that suppresses components shared across conditions.

2. **Trajectory-based visualization.** We treat $\Delta h(t)_{t=1}^{T}$ as an ordered trajectory that directly reflects condition-induced state transitions, indexed by token position and visualized using a deterministic dimensionality reduction method (PCA). This yields a compact representation of token-indexed state transitions under a fixed pipeline.

3. **Cross-model empirical consistency.** Using the same analysis pipeline, we demonstrate that qualitatively similar hidden-state trajectory structures emerge across independently developed Transformer-based LLMs, including models with different architectures, training regimes, and hidden dimensions.

Overall, our method provides a model-agnostic measurement framework for observing token-ordered hidden-state trajectories, and reveals a shared geometric structure in Transformer hidden-state dynamics that is reproducible across independently developed LLMs.

## 2. Related Work

### 2.1. Hidden-State Analysis

A broad line of work investigates the internal representations of neural language models through probing methods and diagnostic classifiers, with the goal of identifying whether particular linguistic, semantic, or task-relevant attributes are encoded in hidden states. Representative approaches include structural probes and layer-wise analyses that correlate hidden representations with syntactic or semantic properties (e.g., (Hewitt & Manning, 2019; Tenney et al., 2019; Belinkov & Glass, 2019; Rogers et al., 2020)). While these methods have been effective at characterizing what information can be linearly recovered from activations, they typically treat hidden states as static snapshots associated with individual tokens or spans. As a result, their primary outputs are scalar scores or classification performance, rather than representations that capture how internal states evolve across a sequence.

Another prominent line of work focuses on attention visualization as a means of interpreting model behavior (e.g., (Clark et al., 2019)). However, several studies caution that attention weights alone may not constitute faithful explanations of model decisions, motivating a careful separation between visualization and causal attribution (e.g., (Jain & Wallace, 2019; Serrano & Smith, 2019)). Further work on pruning and head importance suggests that attention patterns can be redundant or non-essential for output behavior (e.g., (Michel et al., 2019)), complicating interpretability claims based solely on attention maps.

Beyond probes and attention, representation analysis frequently relies on similarity measures and activation statistics. Methods such as SVCCA and CKA compare representations across layers, models, or training checkpoints by summarizing activations as distributions or subspaces (e.g., (Raghu et al., 2017; Kornblith et al., 2019)). Other approaches analyze specific components, such as feed-forward layers, through functional or mechanistic interpretations (e.g., (Geva et al., 2021)), or intervene on internal states to test causal influence on outputs (e.g., (Meng et al., 2022)). While these methods provide important insights into internal structure, they typically emphasize point-wise activations, aggregated summaries, or intervention effects, rather than explicitly constructing and comparing token-ordered state-transition objects.

In a related but distinct direction, several works incorporate hidden states directly into training objectives, particularly in the context of knowledge distillation. For example, Dasgupta & Cohn (2025) propose improving language model distillation by explicitly matching teacher and student hidden states using similarity measures such as CKA, enabling alignment even when the representations differ in dimensionality. These approaches treat hidden states as optimization targets, encouraging representational similarity across models to facilitate knowledge transfer.

While such methods highlight the functional importance of hidden representations during training, their primary focus lies in enforcing correspondence between models rather than in observing or characterizing internal dynamics. In contrast, our work treats hidden states as measurement outcomes rather than loss terms. We do not seek to align or modify representations; instead, we construct a visualization object that directly exposes how hidden states change across token positions under controlled input differences. This separation allows us to study internal state transitions without conflating measurement with optimization.

Recent work has further emphasized the importance of intermediate layers by proposing quantitative frameworks for evaluating representation quality across depth. For example, Skean et al. (2025) introduce information-theoretic and geometric metrics to assess how representations evolve layer by

layer, demonstrating that intermediate layers often outperform final layers across a wide range of downstream tasks. While such approaches provide strong evidence that valuable structure resides within hidden states, they primarily characterize representations through scalar metrics or task performance, rather than through explicit geometric objects tracing token-wise evolution.

Most closely related to our perspective, recent work has proposed hidden-state predictability as a proxy for in-context computational complexity, arguing that next-token loss alone fails to capture the amount of computation performed internally (Herrmann et al., 2025). While this approach treats hidden states as prediction targets to quantify computational difficulty, our work instead treats hidden states as measurement outcomes, constructing a geometric visualization object that directly exposes token-wise state transitions under controlled input differences.

### 2.2. Algorithmic and Automaton Perspectives

A closely related line of research studies how neural networks acquire algorithmic or automaton-like internal structure through training. In particular, van Rossem & Saxe (2025) analyze the learning dynamics of recurrent neural networks on streaming parity tasks by clustering hidden states and constructing discrete finite automaton (DFA) proxy models. They show that training proceeds through distinct phases: an initial expansion in the number of effective states that overfits the training data, followed by a merging phase in which states collapse until a finite automaton emerges, enabling out-of-distribution generalization. Their work combines empirical analysis with simplified dynamical models to explain when and why state merging occurs under gradient flow.

This line of work provides strong theoretical and empirical evidence that continuous hidden-state dynamics can give rise to discrete algorithmic structure. However, its primary objective is to extract and validate discrete abstractions—such as DFA states—that summarize long-term behavior. In contrast, our work does not aim to identify or validate discrete automaton structure. Instead, we focus on constructing a continuous, token-indexed visualization object that directly exposes how internal state differences evolve along the input sequence under controlled condition changes. Our method is therefore complementary: it operates at a finer temporal and geometric resolution, and it remains applicable regardless of whether the underlying dynamics ultimately admit a clean discrete abstraction.

### 2.3. Dimensionality Reduction

Visualizing high-dimensional neural activations generally requires dimensionality reduction. Principal component analysis (PCA) remains a standard baseline because it is linear, deterministic, and reproducible under fixed preprocessing, providing a stable coordinate system for comparison (Hotelling, 1933). Nonlinear techniques such as t-SNE and UMAP are widely used for exploratory visualization and can emphasize local neighborhood structure (van der Maaten & Hinton, 2008; McInnes et al., 2018). However, such embeddings are often sensitive to hyperparameters and optimization details, making cross-condition or cross-trajectory comparison less straightforward when a consistent coordinate system is required.

In this work, we prioritize reproducibility and comparability within each figure. Accordingly, we adopt PCA as a deterministic linear projection applied to pooled differential vectors, ensuring that all trajectories within a figure are expressed in a shared coordinate system. We do not claim PCA to be optimal for all analyses, nor do we preclude alternative projection methods, which are intentionally left outside the scope of this measurement-focused study. Moreover, we find that the same projection procedure yields qualitatively similar differential trajectory structures across independently developed Transformer-based language models. Despite differences in architecture, training data, and hidden dimensionality, the resulting trajectories exhibit consistent geometric organization under identical preprocessing and projection. This cross-model reproducibility supports the interpretation of the visualized trajectories as reflecting shared structural properties of Transformer hidden-state dynamics, rather than artifacts of a specific model or projection choice.

### 2.4. Positioning

Most prior analyses of neural language model internals can be grouped into approaches that (i) evaluate hidden states as points via probes or diagnostics, (ii) visualize attention or other intermediate quantities, or (iii) summarize activations as distributions or subspaces using similarity metrics.

In contrast, our work centers on an explicitly constructed trajectory object: an ordered sequence of per-token differential displacements $\Delta h(t)$ obtained by subtracting aligned hidden states from two runs that differ only in a controlled input condition. This formulation suppresses shared background components while preserving condition-induced changes, and it yields a visualization target—a token-indexed path—that is distinct from snapshot-based, aggregate, or purely discrete analyses. By treating internal state evolution as a measurable geometric object, our approach provides a complementary lens on hidden-state dynamics that is agnostic to specific architectures and independent of downstream semantic interpretation.

# 3. Method: Neural MRI via Differential Analysis

## 3.1. Overview and Differential Construction

Figure 1 illustrates the core operation underlying our method. For a fixed model and an aligned token sequence of length $T$, we record the hidden state at each token position $t$ from two runs that differ only in a controlled input condition.

Throughout this paper, we denote paired runs as conditions A and B. In all experiments, condition B is defined as a length-matched neutral reference input constructed to preserve strict token-wise alignment while removing the lexical content under investigation. For each condition A, the corresponding condition B uses the same token length and structural template, differing only in the replacement of the target lexical content with a neutral placeholder sequence. This design ensures that $\Delta h(t)$ isolates the differential contribution of the controlled input variation without introducing additional semantic assumptions.

Let $h_A(t)$ and $h_B(t)$ denote the recorded hidden states at position $t$ under conditions A and B, respectively. We define the per-position differential hidden state as

$$\Delta h(t) = h_A(t) - h_B(t).$$

This subtraction suppresses components shared across the two conditions, yielding an ordered sequence of displacement vectors attributable to the controlled difference. Throughout this paper, Figure 1 should be read as defining a measurement procedure, not as a claim about what the resulting trajectory *means*.

## 3.2. Hidden State Acquisition

We study autoregressive decoder-only Transformer language models for which token-indexed hidden states can be recorded during a forward pass. All analyses assume deterministic inference settings to ensure reproducibility of hidden-state trajectories.

Let an input be tokenized into a sequence

$$x = (x_1, \ldots, x_T)$$

using a fixed tokenizer. All runs are executed in inference mode with stochastic components disabled (e.g., dropout off), so that identical inputs yield identical internal activations.

To guarantee token-level alignment between conditions, we use teacher forcing: the full token sequence to be analyzed is provided to the model, and hidden states are recorded for the provided tokens rather than for sampled continuations. For a fixed layer $\ell$, we record a vector representation

$$h^{(\ell)}(t) \in \mathbb{R}^d$$

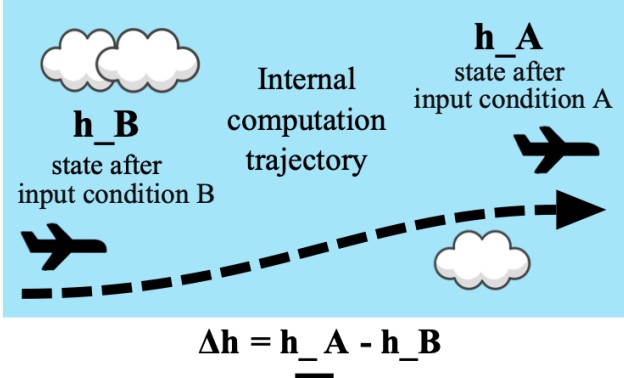

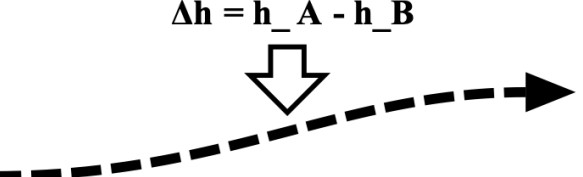

*Figure 1.* Conceptual illustration of the differential hidden-state operation. Hidden states obtained under two aligned input conditions (A and B) are subtracted at each token position to yield a per-position displacement $\Delta h(t)$, suppressing components shared across conditions. Condition B serves as a length-matched neutral reference input for baseline subtraction. The figure specifies the measurement pipeline (state acquisition → differencing → projection → trajectory), not a semantic interpretation of the resulting curve.

at every token position $t \in \{1, \ldots, T\}$, where $d$ is the model hidden size. All results reported in this paper use the same fixed extraction layer $\ell$ across experiments.

In a decoder-only transformer, $h^{(\ell)}(t)$ is extracted at the block output, i.e., after the MLP and residual addition (post-MLP, post-residual), using a fixed checkpoint of the residual stream at layer $\ell$. We hold the extraction checkpoint fixed across all experiments; see §3.6.

Recording over all positions yields an ordered sequence

$$H^{(\ell)} = \{h^{(\ell)}(t)\}_{t=1}^{T}.$$

We retain token identities $x_t$ and indices $t$ so that each trajectory step corresponds to exactly one token position.

When a controlled condition involves a multi-token phrase, we record states for each constituent token under the same one-step-per-token convention. Across all figures, we use the same tokenizer, maximum sequence length, padding and truncation rules, extraction layer $\ell$, and recording protocol to ensure comparability.

## 3.3. Trajectory Formation and Projection

Given two runs A and B that are aligned and identical except for a controlled input condition, we obtain hidden-state sequences

$$H_A^{(\ell)} = \{h_A^{(\ell)}(t)\}_{t=1}^{T}, \quad H_B^{(\ell)} = \{h_B^{(\ell)}(t)\}_{t=1}^{T}.$$

We construct the differential hidden-state sequence

$$\Delta H^{(\ell)} = \{\Delta h^{(\ell)}(t)\}_{t=1}^{T},$$

which is the primary object analyzed in this work.

We use the term *trajectory* in an operational sense: it refers solely to the token-indexed ordering of differential hidden states, not to continuity, smoothness, or dynamical assumptions in a topological or physical sense.

To visualize $\Delta H^{(\ell)}$, we preserve token order and treat the sequence as an ordered trajectory: each $\Delta h^{(\ell)}(t)$ is treated as a point, and successive points are connected to form a polyline indexed by token position. Because $\Delta h^{(\ell)}(t) \in \mathbb{R}^d$ is high-dimensional, we apply a dimensionality-reduction operator

$$\Pi : \mathbb{R}^d \to \mathbb{R}^k, \quad k \in \{2, 3\},$$

to obtain a low-dimensional representation suitable for visualization.

In all experiments, we use principal component analysis (PCA). For each figure, we pool all vectors to be displayed (i.e., all $\Delta h^{(\ell)}(t)$ across all plotted conditions), mean-center them, and fit PCA on this pooled set. Each $\Delta h^{(\ell)}(t)$ is then projected using the same PCA basis, ensuring that comparisons *within a figure* are performed in a shared coordinate system. The resulting plotted trajectory is

$$\{\Pi(\Delta h^{(\ell)}(t))\}_{t=1}^{T},$$

with edges connecting successive token positions.

Importantly, PCA selects directions of maximal variance *in the pooled differential responses* and therefore visualizes the dominant (largest-amplitude) components of condition-induced movement in $\Delta H^{(\ell)}$ under a fixed coordinate system.

PCA is used here as a deterministic linear projection that preserves relative geometry across conditions within each figure. We do not claim PCA to be optimal for all analyses, nor do we exclude alternative projection methods, which are intentionally left outside the scope of this measurement-focused work. Explained-variance ratios can be reported for reference, but are not used as a criterion for interpretation in this study.

### 3.4. Background Suppression and Alignment

The differential operation is designed to suppress components shared across conditions while retaining components that change under the controlled input difference. For each token position $t$, suppose the recorded hidden states can be decomposed as

$$h_A^{(\ell)}(t) = s^{(\ell)}(t) + a^{(\ell)}(t), \quad h_B^{(\ell)}(t) = s^{(\ell)}(t) + b^{(\ell)}(t),$$

where $s^{(\ell)}(t)$ denotes any component shared between runs due to the common model, tokenizer, and shared prompt context at position $t$, and $a^{(\ell)}(t)$ and $b^{(\ell)}(t)$ denote condition-specific residuals. Then

$$\Delta h^{(\ell)}(t) = h_A^{(\ell)}(t) - h_B^{(\ell)}(t) = a^{(\ell)}(t) - b^{(\ell)}(t),$$

so the shared component $s^{(\ell)}(t)$ cancels exactly under subtraction.

Operationally, we maximize the shared component by holding constant all factors other than the designated condition, including model parameters, tokenizer, teacher-forced token positions, extraction layer, and deterministic execution settings. Alignment is critical: if tokenization or sequence length differs between runs, then $\Delta h^{(\ell)}(t)$ would compare states measured at different positions. Accordingly, we design paired conditions to preserve position-wise alignment by construction, using matched-length templates and controlled substitutions that maintain compatible tokenization. The per-position subtraction is applied only when token indices correspond one-to-one between runs.

We treat this alignment requirement not as a limitation of the method, but as a measurement condition necessary for well-defined subtraction. Comparisons across unaligned token positions fall outside the intended scope of this protocol.

### 3.5. Considerations for Long Sequences

When applying this method to long-form inputs, additional design considerations arise. As sequence length increases, issues related to token alignment, accumulation of shared contextual structure, and the geometry of differential trajectories become more prominent. These considerations do not modify the measurement principle defined above, but they affect how trajectories are constructed, segmented, and compared in practice. We summarize and discuss these considerations in Appendix B.

### 3.6. Implementation Details

Unless stated otherwise, we extract $h^{(\ell)}(t)$ from a fixed layer $\ell$ using the same extraction checkpoint across all experiments. All experiments are conducted under deterministic settings with stochastic components disabled. PCA is fitted separately for each figure on the pooled set of vectors to be displayed after mean-centering, and the same PCA basis is used to project every trajectory within that figure. Trajectories are visualized in three dimensions unless otherwise noted. The differential operation is applied only when token indices are aligned one-to-one between runs; cases with tokenization-induced misalignment are excluded from subtraction-based analysis.

Throughout this section, we emphasize that the described pipeline defines a measurement procedure. It does not as-

sign semantic labels, decision boundaries, or functional interpretations to the resulting trajectories. Any downstream interpretation is explicitly external to the measurement itself.

# 4. Visualization

## 4.1. Dimensionality Reduction

For each figure, we pool all vectors that will appear in that plot (e.g., all $\Delta h(t)$ across the shown conditions), mean-center the pooled set, fit PCA on this pooled set, and then project every vector using the same PCA basis. This ensures that all trajectories within a figure share a common coordinate system. Unless noted otherwise, we visualize the first $k$ principal components with $k = 3$ (3D plots). For transparency, we report the explained-variance ratios of the retained components in Appendix A.

We do not apply variance normalization or other feature-wise standardization beyond mean-centering. No additional non-linear post-processing is applied beyond mean-centering and PCA. The projected trajectories therefore reflect the dominant linear variance structure of the included differential vectors under the stated protocol. We emphasize that PCA is fitted per figure; cross-figure comparisons should be made qualitatively under consistent plotting conventions, rather than by assuming a globally shared coordinate system.

## 4.2. Trajectory Construction

After computing per-position differences $\Delta h(t)$ for aligned token indices $t = 1, \ldots, T$, we represent the ordered sequence

$$\Delta H = \{\Delta h(t)\}_{t=1}^{T}$$

as a trajectory.

Each token position corresponds to a single point in the projected space,

$$z(t) = \Pi(\Delta h(t)) \in \mathbb{R}^{k},$$

and consecutive positions are connected to form an ordered polyline. This construction preserves token order: the visualization is a token-indexed path induced by sequential processing. Multi-token expressions yield multi-step trajectories (one step per token), whereas shorter inputs yield correspondingly shorter paths under the same convention.

Within each figure, trajectories share the same PCA basis and axis definitions (principal components), enabling direct within-figure comparison under a common coordinate system.

## 4.3. Figure Overview

Figures 2–5 present differential hidden-state trajectories under increasingly diverse input conditions, ranging from low-ambiguity numeric sequences to related lexical phrases and heterogeneous lexical categories, all visualized under identical preprocessing and projection settings.

# 5. Experimental Observations (Results)

We report differential hidden-state trajectories produced by the fixed pipeline illustrated in Figure 1. For each aligned condition pair, we compute $\Delta h(t) = h_A(t) - h_B(t)$ at corresponding token positions and project the resulting sequence into a shared PCA space. Across all experiments, trajectories are visualized as ordered paths indexed by token position. We report geometric properties of these trajectories without assigning semantic meaning to axes, separations, or regions. All projections shown correspond to the first three principal components (PC1–PC3).

Unless otherwise stated, all results in Figures 2–5 are generated using the Mistral-7B model with the fixed extraction layer $\ell$ described in Section 3.2.

## 5.1. Numeric Token Trajectories

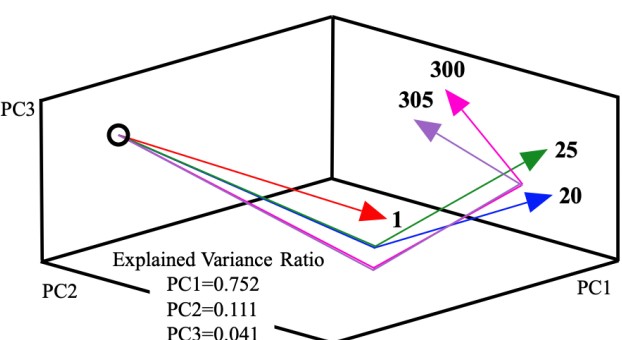

*Figure 2.* Differential hidden-state trajectories for numeric token sequences projected into a PCA space. Trajectories are ordered by token position and visualized using the first three principal components (PC1–PC3), which together explain approximately 90% of the variance.

Figure 2 shows trajectories obtained from numeric token sequences under the same differential pipeline. Numeric inputs provide a low-ambiguity setting in which token order and identity are explicit, serving as a sanity-check domain for sequential state changes. Each appended numeric token corresponds to a successive step along the curve, yielding an ordered trajectory rather than an unordered point cloud. As token index advances, trajectories exhibit coherent progression consistent with token-wise updates captured by $\Delta h(t)$.

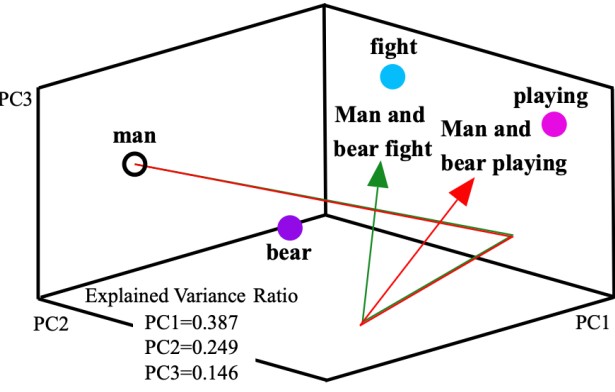

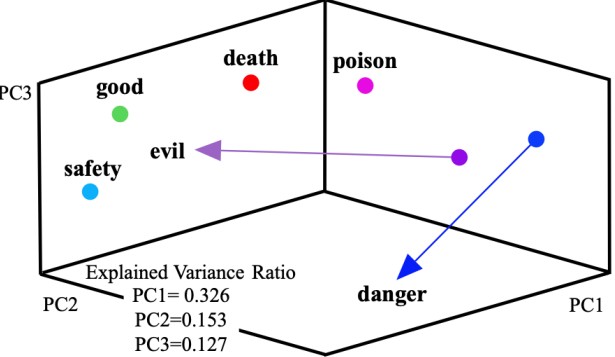

*Figure 3.* Differential hidden-state trajectories for selected water-related lexical inputs, visualized as token-indexed paths in a shared PCA space.

## 5.2. Water-related Lexical Inputs

Figure 3 applies the same procedure to a set of water-related lexical inputs and their constituent elements (e.g., "ice," "water," "boiling water," "H2O," as well as "hydrogen" and "oxygen"). Single-token inputs yield short trajectories, whereas multi-token phrases yield multi-step paths within the same projection. Within the shared space, trajectories trace ordered paths and occupy structured regions across token positions, despite increased lexical variability relative to numeric inputs.

## 5.3. Human and Animal Lexical Relationships

*Figure 4.* Differential hidden-state trajectories for related lexical inputs involving humans and animals. Trajectories show token-indexed displacements $\Delta h(t)$ projected onto PC1–PC3 using PCA (total explained variance 0.782).

Figure 4 presents trajectories for lexical inputs involving humans and animals. Conditions share overlapping tokens and differ by controlled substitutions or modifiers, including single-token inputs ("man," "bear," "fight," "playing") and multi-token expressions ("Man and bear fight," "Man and bear playing"). Closely related phrases yield trajectories that are nearby yet separable within the same projection, while preserving token-indexed ordering for multi-token inputs.

## 5.4. Risk-related and Contrastive Lexical Inputs

*Figure 5.* Differential hidden-state trajectories for contrastive and risk-related lexical inputs projected into a PCA space.

Figure 5 applies the same visualization pipeline to a diverse set of contrastive lexical inputs (e.g., "danger," "death," "poison," "evil," "safety," "good"). These inputs are selected to span opposing or qualitatively distinct lexical categories within a shared projection. Under identical analysis settings, trajectories form distinct paths and occupy different regions, enabling side-by-side comparison of relative placement and trajectory geometry.

## 6. Discussion

Our results show that differential hidden-state trajectories produced by the proposed pipeline exhibit qualitatively similar geometric structure across independently developed Transformer-based language models. Despite differences in architecture, training data, optimization procedures, and hidden dimensionality, the same preprocessing, differencing, and projection steps yield comparable token-ordered trajectories. This cross-model consistency indicates that the observed structures are not artifacts of a particular implementation or projection choice, but instead reflect shared properties of Transformer hidden-state dynamics under controlled input differences.

Importantly, the purpose of this analysis is not to assign semantic meaning to specific directions, regions, or axes in the projected space. The trajectories are treated as measurable geometric objects whose organization can be compared across conditions and models, rather than as representations requiring interpretive labels. In this sense, the method functions as a measurement procedure: it exposes how internal states change in response to controlled perturbations, without presupposing what those changes "mean" in linguistic or cognitive terms.

This perspective distinguishes our approach from many interpretability methods that attempt to explain model behavior by mapping internal activations to human-interpretable concepts. By construction, the differential operation sup-

presses components shared across conditions and highlights condition-induced transitions, yielding a token-indexed path that can be directly observed and compared. The resulting visualization does not constitute an explanation of model decisions, but rather an observation of internal state evolution analogous to a physical measurement that precedes interpretation.

The observation that similar trajectory structures arise across models suggests the presence of common geometric responses induced by the Transformer architecture itself. From this viewpoint, hidden-state trajectories are not idiosyncratic byproducts of training, but reproducible manifestations of how Transformer-based models process sequential input. This supports treating hidden-state dynamics as an empirical object of study, independent of downstream task performance or output-based evaluation.

Overall, the proposed framework reframes hidden-state analysis from an interpretive exercise into a measurement problem. By providing a simple, reproducible procedure for observing token-ordered state transitions under controlled differences, Neural MRI offers a shared observational basis upon which further theoretical or mechanistic analyses can be built.

## 7. Limitations

Our study has three primary limitations.

First, the proposed method requires access to token-indexed hidden states from the target model. Consequently, it cannot be applied directly to closed models that do not expose internal activations, and our empirical evaluation is limited to instrumented or open-weight models. This constraint reflects an instrumentation limitation rather than a conceptual restriction of the measurement procedure itself.

Second, all visualizations rely on dimensionality reduction. Although we employ PCA to obtain a deterministic and reproducible projection, any low-dimensional visualization represents only a projection of an underlying high-dimensional hidden-state sequence. While PCA preserves directions of maximal variance, it may discard lower-variance components that could still be functionally relevant. Accordingly, the geometric relations observed in the figures characterize the projected representation under the stated protocol and should not be interpreted as a complete description of the original hidden-state space. Importantly, the use of PCA affects only visualization and not the underlying measurement, which is defined entirely in the original hidden-state space.

Third, the differential operation assumes a one-to-one alignment of token positions across paired runs. If experimental conditions induce differences in tokenization or sequence length, per-position subtraction becomes ill-defined without additional alignment mechanisms. Addressing such cases requires extensions beyond the current pipeline and is left for future work. In this sense, token alignment should be understood as a measurement condition rather than a modeling assumption.

Taken together, these limitations delineate the practical scope of the proposed measurement procedure and underscore the importance of careful experimental design when constructing aligned condition pairs.

## 8. Conclusion

We introduced Neural MRI, a differential visualization framework designed as a measurement instrument for observing hidden-state trajectories in large language models. Given two aligned input conditions, the method records token-indexed hidden states, computes per-position differences $\Delta h(t) = h_A(t) - h_B(t)$, and renders the resulting ordered sequence as a trajectory using a deterministic PCA projection.

This procedure yields a reproducible measurement object: a token-ordered path of differential state displacements that enables direct comparison across conditions within a shared coordinate system. Using a fixed analysis pipeline and consistent visualization settings, we demonstrated the approach on numeric sequences and multiple lexical input sets, and showed that coherent trajectory structures arise consistently across independently developed Transformer-based models.

Crucially, the emergence of similar differential trajectory geometry across multiple architectures suggests that these patterns reflect shared structural properties of Transformer hidden-state dynamics, rather than artifacts of a specific model or visualization choice. From an observational standpoint, this supports the view that token sequences are internally processed as structured state transitions within a high-dimensional geometric space. Importantly, this claim is grounded in measurement: the method exposes ordered internal transitions directly, without assigning semantic labels or relying on output-based interpretation.

Overall, our results show that condition-induced hidden-state transitions in LLMs can be made observable through simple differencing combined with standardized trajectory-based visualization. The observed cross-model reproducibility further supports the interpretation that LLMs internally process token sequences as structured state transitions in a high-dimensional geometric space. Neural MRI therefore provides a model-agnostic framework for measuring token-wise internal state evolution, offering a reproducible observational basis for future theoretical, mechanistic, and empirical studies of hidden-state dynamics.

## Impact Statement

This work aims to advance the scientific understanding of internal dynamics in large language models by providing a reproducible measurement and visualization framework. The proposed method is non-invasive and does not modify model behavior, reducing risks associated with deployment or misuse. While such tools may inform future model analysis and safety research, we do not anticipate immediate negative societal impacts beyond those already associated with general machine learning research.

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

# Appendix

This appendix provides supplementary material that clarifies scope, assumptions, and boundary conditions of the proposed measurement pipeline.

**Supplementary Video.**
Supplementary Video. A supplementary video (pcafig4.mp4) demonstrates the execution of the PCA-based visualization pipeline, including hidden-state extraction, differencing, and PCA projection, from input specification to interactive trajectory rendering. The video further includes demonstrations of the same pipeline applied sequentially to multiple independently developed Transformer-based language models, illustrating the cross-model reproducibility of the observed differential hidden-state trajectories. Static waiting intervals without screen changes are temporally compressed for viewing clarity. It is organized as follows:

- **Appendix A: Cross-Model Reproducibility Experiments** Additional differential trajectory visualizations demonstrating consistent hidden-state geometry across independently developed Transformer-based language models.

- **Appendix B: Reviewer Q&A** Clarifications addressing anticipated technical questions.

- **Appendix C: Hallucination as Continuity-Preserving Completion** Non-normative commentary.

- **Appendix D: Human Conditioning and Persuasion Effects** Prompt framing effects.

- **Appendix E: An Intuitive View on High-Dimensional Representations and State Space** Optional conceptual perspective.

- **Appendix F: Interpreting Hidden States as Constraint-Satisfied Configurations** Optional interpretive lens on state formation and decoding.

- **Appendix G: Optional Reading Cautions and Technical Boundaries** Interpretation boundaries.

## A. Cross-Model Reproducibility of Differential Trajectories

This appendix reports additional experimental evidence demonstrating that the proposed Neural MRI pipeline produces qualitatively consistent differential hidden-state trajectories across independently developed Transformer-based language models.

Appendix A explicitly replicates the experimental setup used in Figure 4 (Human and Animal Lexical

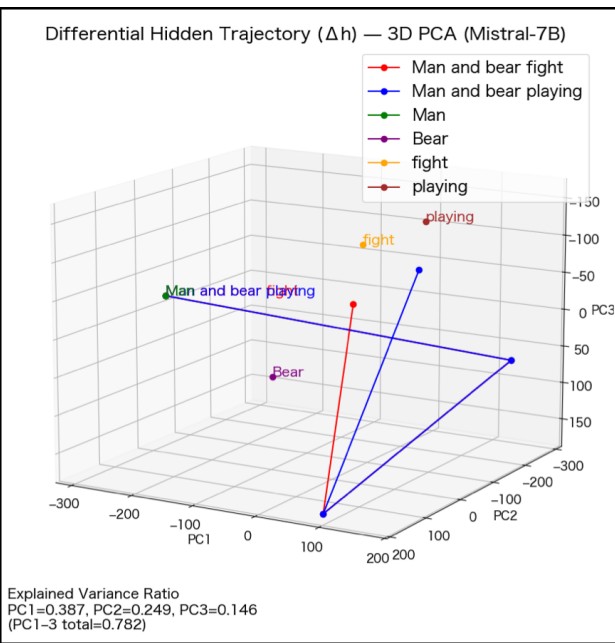

*Figure 6.* Differential hidden-state trajectories for Mistral-7B under the Neural MRI pipeline.

Relationships) across multiple models using an identical differential pipeline and visualization protocol. Figures in the main text apply minor visual simplifications for clarity and layout, whereas the figures in this appendix present unmodified trajectory visualizations generated under the same fixed analysis settings.

The visual simplifications applied to the main-text figures are limited strictly to presentation choices, including font size, line thickness, legend placement, and axis visibility. Importantly, the underlying data, differential hidden-state vectors $\Delta h(t)$, PCA projections, explained variance ratios, and token-indexed trajectories are identical between the main text and appendix figures. No numerical values, projections, or trajectory geometry are altered.

We apply the identical analysis protocol described in Sections 3 and 4 to three representative decoder-only Transformer language models developed by different organizations: Mistral-7B, Phi-3-mini, and Gemma-7B-it. These models differ substantially in architectural details, training data, optimization procedures, and hidden-state dimensionality. No model-specific tuning or modification of the Neural MRI pipeline is introduced beyond standard tokenizer configuration and deterministic inference settings.

All appendix experiments use the same fixed extraction layer $\ell$ specified in Section 3.2. For each model, we record token-indexed hidden states under aligned input conditions, compute per-position differencing

$$\Delta h(t) = h_A(t) - h_B(t),$$

and project the resulting sequences into a three-dimensional

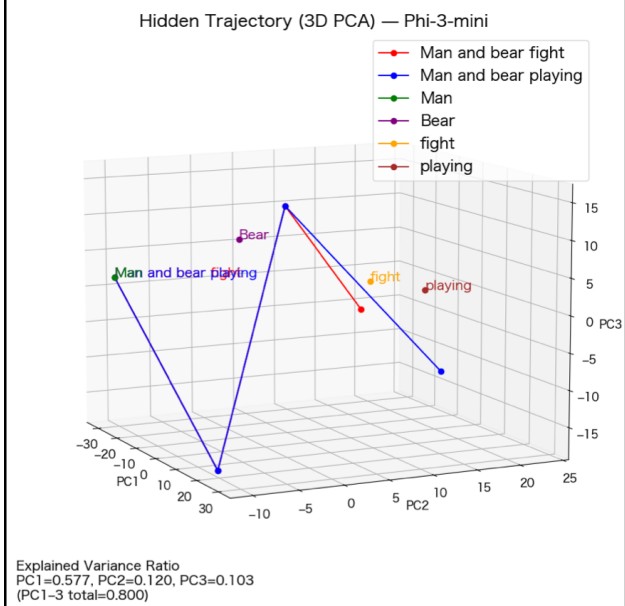

*Figure 7.* Differential hidden-state trajectories for Phi-3-mini under the Neural MRI pipeline.

PCA space using the same pooling, mean-centering, and projection procedure described in the main text. All visualizations are generated using identical plotting conventions.

Figures 6, 7, and 8 present the resulting trajectories for the same set of lexical inputs across the three models. Despite differences in scale and absolute coordinate ranges, the trajectories exhibit comparable geometric organization and token-ordered structure. In particular, related lexical inputs produce coherent, structured paths rather than arbitrary or unstructured scatter, and small input variations induce consistent directional changes within each model.

These results indicate that the observed differential trajectory structures are not artifacts of a specific model implementation or training regime. Instead, they reflect reproducible properties of hidden-state dynamics shared across Transformer-based language models when subjected to the same controlled input differences. This cross-model reproducibility supports interpreting Neural MRI as a model-agnostic measurement procedure rather than a model-specific analysis technique.

## B. Reviewer Q&A

This appendix anticipates technical questions that may arise during review. The main paper presents a measurement pipeline and geometric observations of differential hidden-state trajectories. The items below clarify which statements are empirical observations, which are methodological design choices, and what is explicitly out of scope. The goal is not to extend the main claims, but to

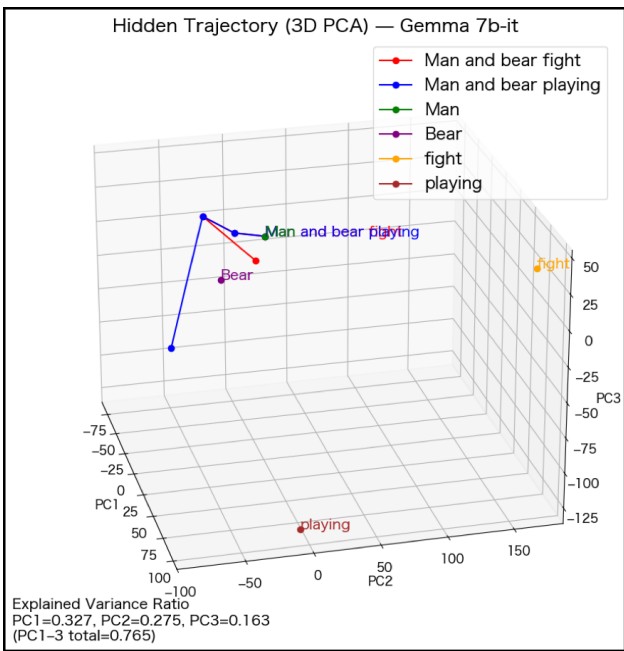

*Figure 8.* Differential hidden-state trajectories for Gemma-7B-it under the Neural MRI pipeline.

make explicit which questions the paper does and does not address.

### B.1. Does the method assume linearity of hidden-state dynamics?

**Q.** The paper uses subtraction of hidden states across conditions. Does this rely on a linear decomposition of hidden states (e.g., "background + content"), and how realistic is that for complex behaviors?

**A.** The method does not assume global linearity of the model's mapping from tokens to hidden states. The core operation is an algebraic difference between two recorded vectors at aligned token positions:

$$\Delta h(t) = h_A(t) - h_B(t).$$

The only linear structure required is that subtraction is well-defined in the vector space where hidden states live. This requirement is agnostic to the specific training history or parameterization of the model, as long as the hidden-state representation is defined in a vector space and recorded consistently.

The cancellation intuition is minimal and procedural: under identical recording settings and token-index alignment, contributions that are identical across the two runs are removed under differencing. This is a statement about the measurement procedure, not a claim that the model is linear or that behavior decomposes additively.

We do not require that $\Delta h(t)$ correspond to a single

"concept vector," nor that hidden-state variation admit a clean additive decomposition. Empirically, we observe that the resulting sequences $\{\Delta h(t)\}$ form stable, token-ordered trajectories under a fixed measurement pipeline. Where differencing yields high variance or unstable geometry under small perturbations, we treat this as an empirical outcome rather than a modeling failure.

### B.2. Could $\Delta h(t)$ be dominated by positional effects or time-step differences?

**Q.** Hidden states depend on position and time step. How do we know the observed geometry is not mainly positional?

**A.** In the main experiments, $\Delta h(t)$ is computed at the same token index $t$ across two runs, using the same tokenizer and aligned token positions via teacher forcing. We do not subtract states from different time steps or different sequence lengths; subtraction is position-wise by construction.

Positional information may still contribute indirectly, but to the extent that positional contributions are identical across aligned runs, they are suppressed by differencing. The limitation is explicit: if conditions change tokenization or length such that alignment breaks, per-position subtraction is not defined without additional alignment procedures. Such cases are excluded from the current pipeline.

### B.3. How do you handle chain-of-thought or long-form generation?

**Q.** If the model produces long answers, internal dynamics may include multiple phases. How does the trajectory view apply to chain-of-thought (CoT) or long-form settings?

**A.** The main paper does not claim to resolve interpretability of long-form reasoning or intermediate reasoning traces. Our strongest results are in settings where aligned token sequences can be defined and $\Delta h(t)$ can be computed reliably.

In long-form generation, trajectories naturally become more complex because the token stream often includes multiple segments, regimes, and topic shifts. The trajectory representation still applies—hidden states can be recorded token-by-token under the same protocol—but downstream analysis becomes more application-specific.

### B.4. Why PCA is used, and how sensitive are results to the projection choice?

**Q.** The figures use PCA projections. Could the observed structure be an artifact of PCA, and would alternative dimensionality reduction methods (e.g., t-SNE, UMAP, or nonlinear embeddings) change the conclusions? Is the choice of PCA a limitation?

**A.** PCA is chosen deliberately as part of the measurement design rather than as an optimization for visual separation. The primary goals of Neural MRI are reproducibility, comparability across conditions, and stability under repeated measurement.

PCA provides a deterministic, linear projection with a well-defined global coordinate system, allowing trajectories from different conditions to be overlaid and compared directly under identical preprocessing. The projection is fit per figure on the pooled set of displayed vectors and applied uniformly; it is not trained to separate classes or optimize visual clustering.

We do not claim that PCA preserves all high-dimensional structure. The correct interpretation is that the figures show the geometry of $\Delta h(t)$ under the stated projection protocol. Alternative embedding methods such as t-SNE or UMAP may reveal local patterns, but they introduce stochasticity, hyperparameter sensitivity, and potential distortions of global geometry. These properties complicate direct comparison across figures, conditions, or models, and are therefore misaligned with the measurement-first objective of this work.

Importantly, the Neural MRI pipeline itself is agnostic to the choice of projection: differential hidden-state trajectories $\Delta h(t)$ are defined and recorded entirely in the original high-dimensional hidden-state space. The use of PCA affects visualization only, not the underlying measurement.

Exploring alternative projections may be appropriate for downstream, application-specific analysis. Such extensions require additional design choices and evaluation criteria, and are intentionally treated as out of scope for the present paper.

### B.5. How reproducible is Neural MRI across models and implementations?

**Q.** Is the method tied to a specific model or codebase? Can other researchers reproduce the main phenomena?

**A.** Reproducibility is a design goal. Neural MRI requires only: (i) a forward pass, (ii) a hook to record token-indexed hidden states at a chosen layer, and (iii) a per-position subtraction between two aligned runs. No fine-tuning, auxiliary training, or optimizer state is involved.

The method is therefore portable across transformer implementations. The critical requirement is alignment: runs must share tokenization, length, and index correspondence. Under deterministic inference, trajectories

are fully determined by the inputs and model weights.

In the main experiments, we verified the stability of differential trajectories across three independently trained transformer-based language models using the same measurement protocol. While we do not claim universality across all architectures or training regimes, this result demonstrates that the observed phenomena are not idiosyncratic to a single model instance.

### B.6. Is Neural MRI intended to automatically compare or evaluate long-form outputs?

**Q.** Long-form answers or stylistic variations make direct comparison difficult. Does this limit the usefulness of Neural MRI?

**A.** Neural MRI is intentionally designed as a measurement and data-logging instrument, not as an automated evaluation, scoring, or decision system. Recording rich internal dynamics necessarily produces complex data, which is expected when observing processes that were previously inaccessible.

Automating comparison or producing task-level scores requires additional analytical layers and assumptions. These belong to downstream tools built on top of the measurement pipeline, not to the definition of the measurement itself.

### B.7. Does the paper make claims about hallucination, refusal, or safety categories?

**Q.** Some readers may want to interpret trajectory geometry in terms of hallucination, refusal, or safety behavior. Are such claims part of the paper?

**A.** No. The main paper does not define or validate behavioral categories such as hallucination or refusal in trajectory space. Such interpretations depend on external labeling protocols, evaluation criteria, and model-specific policies. While differential trajectories may support downstream analysis, these directions require separate experimental design and are intentionally kept outside the scope of the present work.

### B.8. Is the absence of quantitative metrics a limitation?

**Q.** The method does not introduce quantitative metrics. Is this a limitation?

**A.** No. This is a deliberate design choice. Neural MRI is introduced as a measurement and recording instrument for hidden-state transitions, not as an evaluation or decision system. Quantitative metrics depend on task definitions, labeling protocols, and application-specific criteria.

Defining such metrics is a downstream analytical step enabled by this measurement, not a prerequisite for it. This design choice is particularly important in cross-model settings, where defining a single quantitative metric without collapsing model-specific dynamics would require additional assumptions beyond the scope of a measurement-first approach.

### B.9. How should the principal components (PC1–PC3) be interpreted in the visualizations?

**Q.** Some readers may wish to interpret individual principal components (PC1–PC3) as semantic, functional, or safety-related axes. Do the principal components used in this paper carry such meanings?

**A.** No. The principal components shown in the figures are not assigned semantic, functional, or behavioral interpretations. They are treated strictly as orthogonal directions capturing the largest variance in the pooled set of differential hidden-state vectors $\Delta h(t)$ for each figure.

PCA is used solely as a deterministic linear projection to visualize dominant directions of change under controlled input differences. The ordering of components reflects variance magnitude only and does not imply that any specific axis corresponds to meaning, intent, or category. All geometric observations reported in the paper are therefore restricted to relative trajectory structure within the projected subspace, rather than to interpretation of individual axes.

### B.10. Does projecting onto PC1–PC3 discard important information?

**Q.** Since PCA retains only the highest-variance components, could important information contained in lower-variance dimensions be lost, potentially affecting the conclusions?

**A.** Projecting onto PC1–PC3 is intended to reveal dominant differential dynamics, not to preserve the full hidden-state representation. Lower-variance components may encode fine-grained or task-specific effects; however, the claims of this paper are limited to observable geometric structure within the retained subspace.

The explained-variance ratios reported in the supplementary material indicate that a substantial fraction of differential variance lies within a low-dimensional linear subspace for the conditions studied. No conclusions in the paper rely on the interpretation of discarded dimensions, and no claims are made about completeness beyond the measured projection.

### B.11. Do later token positions reflect the influence of earlier tokens?

**Q.** In the presented trajectory visualizations, each point corresponds to a specific token position. Does the hidden state at a later token position (e.g., the second token in a phrase) reflect the influence of earlier tokens, or are token positions treated independently?

**A.** Yes. In autoregressive decoder-only Transformer models, the hidden state at token position $t$ is causally conditioned on all preceding tokens $\{x_1, \ldots, x_t\}$ by design. Through causal self-attention and residual accumulation, information from earlier tokens is propagated and integrated into subsequent hidden states. Accordingly, the differential hidden state $\Delta h(t)$ should be understood as a cumulative state resulting from the entire prefix up to position $t$, rather than as an isolated response to a single token. The resulting trajectories therefore represent ordered state transitions over token positions, not independent pointwise activations.

## C. Hallucination as Continuity-Preserving Completion (Non-normative Commentary)

This appendix provides optional commentary that is consistent with a trajectory-based description of token-indexed state evolution. It introduces no new empirical claims beyond what is measured by the differential trajectory pipeline, and it does not propose diagnostic criteria.

### C.1. Working definition

We use "hallucination" in the common operational sense: generated content that is specific and confident yet unsupported or incorrect with respect to an external reference. Here we give a conservative structural description of how such outputs can arise under weak external constraints.

### C.2. Completion under insufficient constraints

In an autoregressive model, each next-token distribution is produced from a hidden state that integrates the current context. When the context does not sufficiently constrain the space of plausible continuations—e.g., due to missing evidence, underspecified questions, or weak anchoring to verifiable sources—the model can still produce a continuation that is internally coherent in distributional terms. In a state-space description, this corresponds to evolving along a trajectory that remains locally consistent with prompt-imposed constraints, even when those constraints are insufficient to pin down a reference-grounded continuation.

### C.3. Continuity as a sequential bias

A trajectory representation emphasizes that hidden states evolve through incremental updates. Under weak external constraints, a continuation can remain internally consistent while drifting from external correctness. Fluency can remain high because local coherence is achievable even when reference grounding is absent.

### C.4. Scope and cautions

This commentary does not imply that hallucination is reducible to a single geometric signature, nor that it can be diagnosed from visualization alone. It also does not claim that models lack any capacity to represent reference-relevant information. Empirical validation would require explicit labeling protocols and external verifiers, which are outside the scope of the main paper.

## D. Human Conditioning and Persuasion Effects (Non-normative Commentary)

This appendix provides optional commentary consistent with the measurement and trajectory observations in the main paper. It is not required to reproduce the method or figures, and it introduces no new empirical claims.

### D.1. Conditioning by the question: how prompts shape the continuation space

A practical observation in LLM use is that prompt framing constrains the set of plausible continuations. In a trajectory-oriented description, the prompt and surrounding context act as conditions that shape the region of internal states traversed during token generation. Consequently, unusually confident or skewed outputs can be treated as continuations consistent with constraints implied by the user-provided framing, rather than as evidence of a stable, prompt-independent stance.

### D.2. Induced directionality and leading structure

Many prompts implicitly presuppose a preferred conclusion or narrative structure. Structurally, if the prompt establishes a narrow set of acceptable continuations, the model's token-by-token updates are steered toward that subset. This is a statement about conditional generation under constraints; it does not require attributing intent. From this viewpoint, careful prompt design can be understood as careful constraint design, and unsupported outputs can sometimes be traced to underspecified or misleading constraints.

### D.3. Why fluent text can be disproportionately persuasive

LLMs produce text that is locally coherent and stylistically well-formed. Such outputs may be perceived as more reliable than they are because fluent language can compress many unstated assumptions into a polished surface form. In contrast, external checks (e.g., citations, numerical verification, formal derivations, or explicit measurement artifacts) make constraints and failures more visible by forcing intermediate structure to be made explicit.

### D.4. Out-of-domain use and weaker external checkpoints

In-domain users often supply or demand stronger constraints (e.g., required evidence, known failure modes, explicit assumptions). Out of domain, prompts may become underspecified, and coherent continuations can appear without strong anchoring. This can create an asymmetry where the same user experiences reliable behavior in-domain but encounters confident errors out-of-domain.

### D.5. Scope and cautions

These notes do not claim that persuasion, overconfidence, or error can be inferred from trajectory geometry alone. They do not introduce evaluation criteria. Empirically validating these effects would require controlled prompting studies, human evaluation protocols, and external verifiers, which are beyond the scope of the main paper.

## E. An Intuitive View on High-Dimensional Representations and State Space

### E.1. Motivation

The main body of this paper does not rely on any particular intuitive interpretation of high-dimensional representations. All results follow directly from the proposed measurement procedure and empirical observations.

However, during preliminary discussions, we found that misunderstandings often arise from how "dimensions" and "latent space" are intuitively imagined. In particular, it is common to implicitly interpret high-dimensional spaces as extensions of Euclidean coordinates (e.g., many independent $x$–$y$–$z$ axes), which can obscure the nature of the observed structures.

This appendix provides an optional conceptual viewpoint that may help readers build intuition about why the proposed visualization behaves as observed. This perspective is not required for applying the method and does not introduce additional assumptions about model internals or learning dynamics.

### E.2. Dimensions as Attributes Rather Than Coordinates

In many discussions of neural representations, the term *dimension* is used in a strictly mathematical sense, referring to the number of components in a vector space. While correct, this usage can lead to misleading mental models.

An alternative intuition is to regard each dimension not as a spatial axis, but as an attribute contributing to the overall state of a token representation. These attributes are not meant to correspond to explicit variables or disentangled factors.

Examples of such attributes may include (non-exhaustively):

- semantic relevance,

- syntactic role,

- contextual compatibility,

- activation strength,

- temporal dependency,

- interaction potential with other tokens.

From this viewpoint, a token embedding represents a state defined by a large collection of attributes, rather than a point in an abstract coordinate grid.

### E.3. State Space and Global Configuration

Under this interpretation, the hidden state of a model at a given layer can be viewed as a global configuration of states, where:

- each token occupies a position determined by its attribute values,

- the overall geometry reflects relational structure rather than absolute coordinates,

- changes in input induce continuous transformations of this configuration.

Importantly, the structure of interest is not the value of any single attribute, but the collective organization of many attributes across tokens. This helps explain why local, neuron-level inspection often fails to capture meaningful behavior, while global methods reveal coherent structure.

### E.4. Implications for Dimensionality Reduction

This perspective also clarifies why linear dimensionality reduction methods (e.g., PCA) can produce informative projections despite extreme dimensionality reduction.

If the dominant variations in the hidden state correspond to collective shifts in attribute configurations, then:

- principal components naturally align with these global modes,

- projected trajectories reflect transitions between macroscopic states,

- fine-grained attribute interactions are preserved implicitly through correlation structure.

In this sense, dimensionality reduction does not necessarily discard information arbitrarily, but emphasizes dominant patterns already present in the state space.

### E.5. Relation to Dynamic Processing

Because token representations evolve across layers, the state space is inherently dynamic. Each layer transformation can be interpreted as modifying attribute balances rather than relocating tokens in a fixed coordinate system.

From this viewpoint, the observed trajectories correspond to:

- progressive refinement of attribute emphasis,

- stabilization of relational structure,

- convergence toward output-consistent configurations.

This interpretation remains fully consistent with the purely computational view of transformer models.

### E.6. Scope and Limitations

We emphasize that this appendix provides a conceptual aid, not a formal model. In particular:

- no claim is made that individual dimensions correspond to specific semantic attributes,

- no assumption is made about human cognition or biological analogy,

- the visualization results of this paper do not depend on this interpretation.

Its sole purpose is to offer an alternative intuition that aligns naturally with the empirical observations reported in the main text.

### E.7. Summary

Viewing high-dimensional representations as attribute-defined state spaces, rather than extended coordinate systems, may help explain:

- why global visualization reveals coherent structure,

- why local neuron-level interpretation is insufficient,

- why simple linear projections remain informative.

Readers are free to adopt or ignore this perspective without affecting the validity of the proposed method.

## F. Interpreting Hidden States as Constraint-Satisfied Configurations

### F.1. Motivation

The main text of this paper makes no assumptions about how hidden states should be interpreted. All claims rely solely on the proposed measurement procedure and empirical observations.

Nevertheless, we found that discussions around hidden-state behavior are often complicated by implicit linguistic interpretations. In particular, it is common to reason about model behavior in terms of stepwise token generation, which can obscure the fact that the internal computation is fundamentally state-based.

This appendix provides an optional conceptual lens for interpreting the observed trajectories. It does not introduce new claims, results, or assumptions, and is not required for applying the proposed method.

### F.2. State Formation Versus Decoding

From a computational perspective, a transformer model maps an input sequence to an internal hidden state before producing any output tokens. The decoding step then projects this internal state into a probability distribution over the vocabulary.

Under this view, the output sequence should be understood as a linearized readout of an already-formed internal configuration. The hidden state itself represents a constraint-satisfied configuration induced by the input prompt, rather than a sequence of intermediate symbolic steps.

This distinction helps clarify why different prompts that are semantically equivalent can produce similar hidden-state trajectories, even when surface-level token sequences differ.

### F.3. Constraint Satisfaction in State Space

A prompt can be viewed as imposing a set of constraints on the model's internal representation. These constraints may arise from syntax, semantics, contextual dependencies, or task structure.

During the forward pass, the model transforms the hidden state so that these constraints become mutually consistent. Once such a configuration is reached, decoding produces tokens that reflect this state.

In this interpretation, the internal state does not "compute an answer step by step" in a symbolic sense. Instead, it converges toward a configuration from which the output is directly readable under the model's decoding mechanism.

### F.4. Why Linguistic Interpretations Can Be Misleading

Natural language is inherently sequential, which can encourage interpretations that overemphasize token-by-token reasoning. However, hidden-state dynamics operate on the entire representation simultaneously.

As a result, attempting to explain internal behavior purely through linguistic narratives may introduce unnecessary complexity. The apparent reasoning process observed in output text reflects the structure imposed by decoding, not necessarily the structure of the underlying state transitions.

This distinction becomes especially important when interpreting visualization results, which capture state-space motion rather than output generation.

### F.5. Relation to Visualization

The trajectories visualized in this work represent transitions between hidden states across layers. They do not visualize token probabilities or decoded text, but rather the evolution of the internal configuration itself.

Dimensionality reduction methods such as PCA emphasize dominant modes of variation in this state space. From the perspective described here, these modes correspond to global shifts in constraint satisfaction rather than local symbolic operations.

This explains why coherent trajectories can be observed despite aggressive dimensionality reduction.

### F.6. Scope and Limitations

We emphasize that this appendix provides a conceptual aid only.

- No claim is made that hidden states correspond to explicit symbolic representations.

- No assumptions are made about human cognition or biological analogy.

- The proposed measurement method does not depend on this interpretation.

Readers are free to adopt or ignore this perspective without affecting the validity of the results.

### F.7. Summary

Viewing hidden states as constraint-satisfied configurations, rather than sequences of symbolic reasoning steps, may help clarify:

- why visualization captures meaningful structure,

- why decoding can appear simpler than expected,

- and why state-space trajectories remain stable across prompt variations.

This interpretation is offered solely to support intuitive understanding of the observed phenomena.

## G. Optional Reading Cautions and Technical Boundaries (Optional)

This appendix provides optional, non-normative guidance for reading differential trajectory visualizations produced by Neural MRI. It is not required to reproduce the method or figures, and it introduces no new empirical claims beyond those established in the main paper. The main contribution remains a measurement pipeline and geometric observations of differential hidden-state trajectories under aligned conditions.

Throughout this appendix, we avoid semantic attribution and do not define behavioral categories in trajectory space.

### G.1. Why differencing can serve as a primary observable

In many measurement settings, differences are more robust than absolute values because they can reduce shared offsets or common-mode contributions under matched instrumentation. Neural MRI applies this principle to model internals by comparing aligned hidden states under controlled conditions and treating the per-token difference

$$\Delta h(t) = h_A(t) - h_B(t)$$

as the primary measured object, visualized as an ordered trajectory after projection.

This motivates the measurement design; it is not a claim that differencing uniquely identifies causes or concepts.

### G.2. Differential trajectories as measurement outputs

Neural MRI produces token-indexed sequences of vectors derived from recorded activations under a fixed pipeline. These sequences can be treated as measurement outputs that support controlled comparisons across conditions, even when individual coordinates are not directly interpretable. This framing emphasizes observability and reproducibility rather than explanation.

### G.3. Net effect vs. ordered path

A difference can be summarized as an endpoint displacement or studied as an ordered sequence across token positions. Neural MRI uses the second: $\{\Delta h(t)\}$ preserves token order and supports comparisons of how internal states evolve across aligned runs. This does not imply reconstruction of a unique internal mechanism; it provides a consistent object for comparison.

### G.4. Prompts as constraints (descriptive, not mechanistic)

Changing a prompt can change the conditional distribution over continuations and, correspondingly, the internal state evolution during decoding. Differential trajectories provide a measurement object for comparing such changes under controlled alignment, without requiring that any axis correspond to a human-interpretable factor.

This statement is descriptive of conditional generation and the measurement setup, not a mechanistic claim.

### G.5. States as the measured object; text as the interface

The pipeline measures token-indexed hidden states. Generated text is an interface-level artifact of decoding; it is not equivalent to the measured internal dynamics. Neural MRI is designed to complement output-based evaluation by making a class of internal transitions observable under controlled conditions, without asserting semantic interpretation of those transitions.

### G.6. Magnitude and stability (non-binding heuristics)

In practice, one may inspect whether differential trajectories exhibit stable patterns across small perturbations (e.g., minor prompt edits, seed changes under stochastic decoding, or nearby layer choices). Such stability checks can help distinguish robust measurement patterns from projection- or perturbation-sensitive effects. Neural MRI does not define a universal metric here; these are inspection heuristics only.

### G.7. Longer outputs: loops, waypoints, and revisits (descriptive language only)

In long-form setting, projected trajectories may appear more complex (e.g., bends, revisits, or partial loops). Such shapes can be used as descriptive handles for inspection (e.g., "this span shows a geometry shift"), but they should not be treated as evidence of specific internal operations. Projection artifacts and pipeline choices can produce similar visible effects.

### G.8. Multi-scale structure: coarse trends vs. local variation

Trajectory data can be examined at multiple granularities: local segments may vary with phrasing while broader patterns may be more stable across related conditions. The pipeline is compatible with multi-scale inspection (segment-level vs. whole-sequence geometry) without requiring semantic labeling.

### G.9. Avoid semantic substitution

Differential trajectory geometry is not a substitute for correctness checks, citations, verification, or task-level evaluation. A visualization can be informative as a measurement artifact while remaining insufficient for semantic conclusions.

### G.10. Analyst modes: label-first vs. measurement-first

Some analyses begin by assigning semantic labels and then seeking correlates; others begin with stable measured differences and postpone labeling to downstream validation. Neural MRI is optimized for the second: producing comparable measured objects across conditions under alignment. This does not rank interpretive styles; it clarifies the intended use of the pipeline.

### G.11. Difference as a unifying measurement lens (not a law)

Differencing is a practical measurement strategy that can reduce shared contributions under matched conditions. It should not be treated as a universal principle that guarantees interpretability or causal identification. Neural MRI provides a reproducible measured object; stronger claims require additional evidence and task-specific validation.

### G.12. Trajectories are projections of high-dimensional dynamics

Neural MRI visualizes projected trajectories derived from very high-dimensional vectors. The displayed paths preserve token order under the projection protocol but necessarily discard information. Accordingly, the plots

should be read as partial views of internal dynamics, conditioned on the stated layer choice and projection procedure.

### G.13. Tokens as anchors for post hoc observables

The pipeline records hidden states indexed by realized token positions. This provides a stable basis for post hoc comparison across aligned runs. Claims about internal dynamics that do not correspond to token-indexed observables (e.g., transient intermediate computation prior to sampling) require different instrumentation and are treated separately.

### G.14. Notes on "hallucination" framing (boundary statement)

If readers consider downstream analyses related to unsupported continuations ("hallucination"), Neural MRI does not define or detect such categories in trajectory space. Any association between trajectory geometry and behavioral categories requires: (i) external references, (ii) labeling protocols, and (iii) validation experiments. Without these, visual patterns should be treated only as measured differences under the specified conditions.

### G.15. Underconstrained settings and apparent confidence (boundary statement)

In settings where external constraints and verifiable references are weak (e.g., speculative questions), coherent text can be generated without reliable grounding. Neural MRI does not resolve this issue; it makes internal transitions observable under controlled comparisons. Confidence and fluency are not treated as reliability indicators within this work.

### G.16. Measurement before automation

Neural MRI is intentionally limited to measurement and visualization. It is not an automated diagnostic system and does not assign semantic labels, risk scores, or quality metrics. Building automated tools on top of the measured trajectories is a downstream direction that requires additional modeling assumptions, objective definitions, and validation.

### G.17. Boundary of claims: observation without attribution

Neural MRI observes differences in token-indexed hidden-state evolution under aligned conditions. It does not attribute causes, intentions, meanings, or cognitive properties to those differences. Any stronger attribution must be supported by evidence beyond the scope of this paper.

### G.18. Token generation as state transition (descriptive summary)

In autoregressive decoding, each token position corresponds to an update in internal state that conditions the next-token distribution. Neural MRI visualizations summarize these token-indexed updates as an ordered path after projection. This is a description of the measured object (token-indexed state evolution) and does not imply that the trajectory is a literal representation of reasoning or meaning.

### G.19. Why generative pre-emission dynamics are not included

#### G.19.1. OBSERVABLE HIDDEN STATES AND TOKEN-INDEXED REPRESENTATION

This work analyzes internal representations corresponding to realized tokens. These token-indexed hidden states are materialized during execution, can be retrieved after generation, and provide stable objects for post hoc comparison under aligned conditions.

#### G.19.2. TRANSIENT NATURE OF PRE-EMISSION DECODING DYNAMICS

Prior to token emission, internal computation supports candidate evaluation and probability assignment. Intermediate dynamics during this process are not preserved as stable token-indexed objects after sampling unless explicitly captured during decoding.

#### G.19.3. PRINCIPLE OF NON-RETROACTIVE OBSERVABILITY

Once a token is sampled, intermediate computation that contributed to the sampling decision is not, by default, reconstructible from the final output or from post hoc activation snapshots. Recovering such dynamics would require real-time capture during decoding and a different experimental protocol.

#### G.19.4. MEASUREMENT VERSUS INTERVENTION (METHODOLOGICAL SEPARATION)

Capturing pre-emission dynamics typically requires deeper instrumentation of the decoding loop. Depending on implementation, this can alter execution behavior or introduce additional dependencies. This work therefore separates post hoc measurement of realized token-indexed states (non-invasive under typical logging) from real-time decoding instrumentation.

### G.19.5. SCOPE OF THE PRESENT WORK

Accordingly, this work focuses on post-generation, token-indexed hidden-state differences derived from realized tokens. Real-time analysis of pre-emission decoding dynamics is an important direction but requires distinct instrumentation and evaluation assumptions and is left to future work.

### G.19.6. PARALLEL FORMATION OF OUTPUT TENDENCIES (BOUNDARY CLARIFICATION)

During prompt processing, internal states can influence subsequent decoding by shaping the next-token distribution. However, there is no well-defined post hoc observable corresponding to "the hidden state of a future token before it is generated," because token-indexed hidden states are defined for realized token positions. Neural MRI therefore restricts analysis to reproducible, token-indexed observables and avoids retrospective reconstruction of pre-emission dynamics.

## G.20. Measurement, Geometry, and Interpretation

The framework presented in this paper is intentionally limited to the construction of a measurable object derived from recorded hidden states. The resulting trajectories should be understood as geometric artifacts of a differential measurement procedure, rather than as direct representations of semantic meaning or cognitive variables.

Language models operate in high-dimensional representation spaces, where many distinct analytical lenses may be applied after measurement. The role of Neural MRI is to expose a structured, reproducible object in this space—an ordered sequence of differential displacements—while leaving interpretive, functional, or normative analysis explicitly outside the scope of the method.

We view this separation between measurement and interpretation as a prerequisite for systematic study of internal model dynamics, analogous to the role of instrumentation in other empirical sciences.

## G.21. Intuition on GPU-Based Computation and Hidden-State Dynamics

This appendix provides an intuitive perspective intended to aid interpretation of the proposed measurement framework. The discussion below is non-normative and does not introduce new claims or experimental results; rather, it offers a conceptual lens for understanding why differential hidden-state trajectories become observable in large language models.

A well-known empirical fact in the field is that large language models began to exhibit qualitatively different behavior once training and inference were moved to GPU-based architectures. This transition is often attributed to increased computational throughput, parallelism, or scale. While these factors are undoubtedly necessary, they are not sufficient to explain the qualitative shift in internal behavior that motivates hidden-state analysis.

A key distinction lies in the computational substrate itself. CPU-centric computation is naturally described in terms of sequential instruction execution and control flow, whereas GPU-centric computation operates on large collections of vectors and matrices that are updated simultaneously under shared rules. From this perspective, hidden states are not merely intermediate variables in an algorithmic pipeline, but points in a high-dimensional state space that evolve under a fixed update rule.

When a token is processed by the model, it can be viewed as introducing a structured perturbation into this state space. The subsequent forward pass induces a global transformation of the hidden representation, analogous to how placing an object into a physical system induces a change in the surrounding field. Under sufficient scale and resolution, such transformations give rise to coherent trajectories rather than isolated or noisy state changes.

Our method does not assume that these trajectories correspond to semantic meaning, reasoning steps, or human-interpretable concepts. Instead, it treats them as observable consequences of state-space dynamics. By subtracting aligned hidden states across controlled input conditions, shared background components are suppressed, leaving an ordered sequence of displacement vectors that reflects how the system responds to a specific perturbation.

This perspective helps clarify why the resulting trajectories can be visualized without appealing to symbolic reasoning or explicit algorithms. The visualization should be understood as a measurement of state-space motion, not as an explanation of what the model "understands." In this sense, the proposed framework is closer in spirit to diagnostic tools in physics or signal processing than to interpretive or semantic analyses.

Importantly, this intuition does not replace the formal definition of the method. All claims in the main paper are fully specified by the measurement protocol itself. The purpose of this appendix is solely to provide additional context for readers seeking an intuitive understanding of why differential hidden-state trajectories become visible under large-scale, GPU-based computation.

### G.22. On Linguistic Output and Internal State Dynamics

When interpreting the results presented in this paper, it is important to distinguish between *linguistic output* and the model's *internal state dynamics*.

Natural language is inherently sequential and symbolic. As a result, there is a strong tendency to interpret model behavior in terms of token-by-token reasoning, symbolic manipulation, or explicit logical steps. While such narratives may be useful at the level of output analysis, they can be misleading when applied directly to hidden-state behavior.

The measurements and visualizations reported in this work suggest that the primary computational process occurs within a high-dimensional state space, where transformations operate on the entire representation simultaneously. From this perspective, language generation should be understood as a *projection* or *decoding* of an already-formed internal configuration, rather than as a faithful trace of the internal computation itself.

Conflating linguistic structure with internal dynamics may therefore obscure the geometric and dynamical properties observed in hidden-state trajectories. In particular, coherent motion in state space should not be assumed to correspond to discrete symbolic operations or explicit reasoning steps expressed in language.

This paper does not claim that internal states possess semantic meaning in a human-interpretable sense, nor that they correspond to explicit concepts. Rather, the goal of the proposed measurement pipeline is to observe state-space behavior directly, without imposing linguistic or semantic interpretations.

Readers are encouraged to treat linguistic explanations as post hoc descriptions of decoded output, and to interpret the visualized trajectories as properties of the underlying state dynamics themselves.

