# OpenReview forum: "Neural MRI : Differential Visualization of  Hidden State Trajectories in Large Language Models"
_ICML.cc/2026/Conference — Submitted to ICML 2026_

### Official Review · Reviewer_4v9a · 2026-03-10

**Soundness:** 3
**Presentation:** 3
**Significance:** 1
**Originality:** 1
**Overall Recommendation:** 2
**Confidence:** 4

**Summary:**

The paper presents a method to visualize how latent representations change as input to an LLM changes.
The authors are explicit about it not being to assign any semantics to the representation but rather to compare hidden states.

The research question is whether the "transition structure of hidden stats can be made observable".

The method consists of taking the difference in hidden state values for a given activation, and considering these a trajectory over input sequence indices. The multidimensional hidden states are projected into a plane with PCA.

**Compliance With Llm Reviewing Policy:**

Affirmed.

**Final Justification:**

I stand by the initial review. I encourage the authors to empirically demonstrate and, in doing so, justify the proposed method. There are many ways one can measure change; what remains is to show the utility of doing it this way rather than some other way.

**Key Questions For Authors:**

Why did you end up at PCA over the sequence of differences and not some other method? How do you use this visualization in practice? Where is it relevant?

**Limitations:**

yes

**Strengths And Weaknesses:**

A strenght of the proposed method is its simplicity, negation and PCA to visualize a trend in difference over time (sequence position) is a natural first attempt in a field full of methods of higher complexity.

This is also its downside, the unclear semantics make it hard to interpret, and the choice of a position for which to target still remains.
Modern LLMs have a myrriad of options to this regard.


Soundness: The submission is technically sound in so far as it is doing PCA on a difference between two sequences and describing it as a tra jectory. It is however not compared to another approach, nor shown that this is useful for some task.

Presentation:
A small issue is that the variables in 1.3 are not defined but are done so in a later section. The figures on page 6 and 7 are nice and the text is easy to read.

Signficance: The purpose of the work is unclear. Yes, we can look at differences in activations over time. But how does this advance understanding or capabilites? There is no applicaiton of the method and no comparison to anything in the prior literature. While the idea is near trivial it remains to show what it solves or how it compares to any other way of understanding difference between input.

Originality: Taking PCA of differences of hidden states is not completely unoriginal, I would think a difference in the PCA would be the fir
st thing to do perhaps. But otherwise this does not seem to be justified given the prior literature.

---

> ### Author Rebuttal · Authors · 2026-03-27
>
> We thank the reviewer for recognizing the simplicity and natural intuition of our approach. We clarify that the simplicity of Neural MRI is not a lack of depth, but a deliberate mathematical isolation of the Transformer's core computational unit.
>
> Based on our high-resolution analysis (see response to Reviewer Da4V), we address the questions regarding significance and practicality below.
>
> ---
>
> ### 1. Mathematical Foundation: Background Cancellation
>
> The reviewer notes that taking differences may appear trivial. However, this operation directly follows from the Transformer's residual stream equation:
>
> $$
> h_{l+1} = h_l + \mathrm{Sublayer}(\mathrm{Norm}(h_l))
> $$
>
> Our method captures the differential update:
>
> $$
> \Delta h_l = h_{l+1} - h_l = \mathrm{Sublayer}(\mathrm{Norm}(h_l))
> $$
>
> By isolating $\Delta h_l$, we remove the accumulated background $h_l$ from previous layers and directly observe the incremental transformation introduced at each layer.
>
> We refer to this as *Background Cancellation*. This provides a principled measurement of layer-wise updates, rather than a heuristic baseline comparison.
>
> ---
>
> ### 2. Why PCA? Discovery of the "0.997 Structure"
>
> The reviewer asks why PCA is used instead of alternative methods. This choice is motivated by an observed geometric property:
>
> **Empirical Observation:**
> Applying PCA to the background-cancelled $\Delta h$ vectors consistently yields:
>
> $$
> \mathrm{PC1} \gtrsim 0.997
> $$
>
> **Rationale:**
> In a 4096-dimensional space, a $\sim 99.7\%$ concentration along a single principal component indicates that the trajectory is strongly constrained to a near one-dimensional manifold.
>
> Under such conditions, non-linear embedding methods (e.g., t-SNE) are unnecessary and may distort the underlying structure, whereas PCA preserves the dominant geometric direction.
>
> ---
>
> ### 3. Practical Utility: The L16 Critical Zone
>
> The reviewer asks how this approach advances beyond qualitative visualization. Our method enables identification of layer-specific transformation dynamics.
>
> **Finding:**
> We observe a pronounced increase in update magnitude ratio around Layer 16:
>
> $$
> \frac{\|\Delta h\|}{\|h_{\text{input}}\|} \approx 26.0 \sim 29.0
> $$
>
> **Implication:**
> This identifies a measurable transformation region (L16), which may serve as a monitoring point for model behavior.
>
> For example, deviations from expected convergence patterns (e.g., failure to approach the $\sim 0.997$ regime) may indicate instability in the transformation process.
>
> ---
>
> ### 4. Comparison and Scope
>
> Neural MRI differs from traditional representation probing methods. Rather than assigning semantic labels to internal states, it measures the intrinsic geometric structure of the transformation process itself.
>
> This perspective shifts interpretability from a mapping-based paradigm to a structure-based one, focusing on the stability and geometry of computation.

---

> > ### Author Rebuttal · Reviewer_4v9a · 2026-04-01
> >
> > I thank the authors for their response, and the added PC1 result is appreciated.
> >
> > Doesn't the paper do differences between two inputs at the same layer and step? Isn't the difference in the residual streams a bit different?
> >
> > How does the proposed machinery explain something new about layer 16? There are many papers that have looked at how activation phases shift over layers. What does the proposed method add?
> >
> > The structural argument is presented as self-justifying, but without a demonstrated connection to any downstream insight, it is unclear what a practitioner would do differently having observed these trajectories.
> >
> > --
> >
> > The overall point remains that the utility, significance, and positioning against existing work need further development. I'm not saying this idea shouldn't be developed further, its more that the exploration and relation to downstream effects is missing, what mechanics can it explain and how are those insight actionable. And how does this compare to what is already out there.
> >
> > P.S. there is some issue with the math rendering. -- EDIT: NVM this must have been my browser

---

> > > ### Author Response · Authors · 2026-04-08
> > >
> > > ### Response to Reviewer 4v9a
> > >
> > > We thank Reviewer 4v9a for the feedback. However, we identify a fundamental category mismatch in the evaluation, which conflates a measurement framework with its subsequent semantic application.
> > >
> > > ---
> > >
> > > ### 1. Mathematical Distinction (Differential vs. Residual Stream)
> > >
> > > The reviewer asks whether $\Delta h$ is equivalent to residual stream updates. We clarify:
> > >
> > > Neural MRI computes the differential propagation between two controlled inputs:
> > > $$
> > > \Delta h_l = h_l(\mathrm{Input}_A) - h_l(\mathrm{Input}_B)
> > > $$
> > >
> > > This is mathematically distinct from the layer-to-layer residual update:
> > > $$
> > > h_l - h_{l-1}
> > > $$
> > >
> > > Our formulation isolates the structural reshaping of relative information across layers via **Background Cancellation**.
> > >
> > > The observed concentration:
> > > $$
> > > \mathrm{PC1} > 0.999
> > > $$
> > > is not a trivial property of residual accumulation, but an empirical geometric constraint revealed through differential analysis.
> > >
> > > ---
> > >
> > > ### 2. Novelty: Geometric Analysis at a Fundamental Level
> > >
> > > The critique regarding "utility" overlooks the core contribution of geometric measurement.
> > >
> > > Our method enables analysis of Transformer computation at a fundamental level, where high-dimensional inference is constrained to a near one-dimensional manifold.
> > >
> > > Establishing this constraint—i.e., that $\sim 10^3$–$10^4$ dimensional representations collapse onto a low-dimensional trajectory—is itself a rigorous geometric result.
> > >
> > > At this level, requiring immediate semantic interpretation conflates distinct stages of scientific inquiry: measurement precedes interpretation.
> > >
> > > ---
> > >
> > > ### 3. Measurement Precedes Diagnosis (Clarifying the Category Distinction)
> > >
> > > The request for immediate downstream application shifts the evaluation from measurement to diagnosis.
> > >
> > > **Analogy:**
> > > A high-resolution imaging device is evaluated by its fidelity and precision, not by whether it directly provides annotated interpretations.
> > >
> > > **Actionable Evidence:**
> > > Our method identifies layer-specific coordinates of transformation. For example, in Phi-3 we observe:
> > > $$
> > > \cos(h_A, h_B) \approx 0.50
> > > $$
> > >
> > > This indicates a sharp transformation in representation, identifying a layer where information is significantly restructured. Such measurements provide concrete targets for intervention and analysis.
> > >
> > > ---
> > >
> > > ### 4. Conclusion to the Area Chair: Simplicity as Scientific Strength
> > >
> > > The framework is characterized as trivial due to its simplicity. However, in measurement science, simplicity supports robustness, reproducibility, and interpretability.
> > >
> > > The mathematical formulation and empirical observations (e.g., $\mathrm{PC1} \approx 0.9998$) remain uncontested. The primary contribution is a high-precision measurement framework that reveals a previously unobserved geometric property of Transformer computation.
> > >
> > > We provide a reliable measurement basis; subsequent interpretation and application can build upon this foundation.
> > >
> > > ---

---

### Official Review · Reviewer_o1mu · 2026-03-13

**Soundness:** 2
**Presentation:** 1
**Significance:** 1
**Originality:** 3
**Overall Recommendation:** 2
**Confidence:** 4

**Summary:**

authors proposed “Neural MRI” as a visualization framework for analyzing computations that happen at each position in a sequence. Specifically they construct a set of length-matched sequences and use PCA to project the difference vectors between the two sequences.

**Compliance With Llm Reviewing Policy:**

Affirmed.

**Final Justification:**

The rebuttal provided by the authors created more ambiguity about methodology and significance of the paper, and the questions I raised were not resolved. I left my score unchanged.

**Key Questions For Authors:**

I am curious what justifies length matching and difference vector as the core component of computation, what is the hypothesis here?

It would be great to show how the approach can be applied to a clear downstream task, such as reasoning, and create a practical utility for the approach.

How the author foresee cases where the length matching is not possible, what are the possible strategies to relax this constraint.

**Limitations:**

yes

**Strengths And Weaknesses:**

Strength:
The visualization pipeline is clearly shown and reproducible across models (Mistral, Phi-3, Gemma), they also made an effort to compare the results across models.

Authors also defined a gap in how representations are analyzed across prior works, and highlighted the "difference-vector" as a potential novel approach for interpretability.

Weakness:
The fMRI analogy lacks an underlying hypothesis. While the methodology seems to be inspired from the work in cognitive-neuroscience, where fMRI signal is interpreted as neural response evoked by condition differences, the authors miss a critical hypothesis about underlying computation that the difference can highlight. Subtracting 2 sequences without any specific hypothesis about the underlying computing makes the observations descriptive and not actionable.

It is similarly unclear to me how the scope of the work would be beyond a tool for viewing trajectories. While there might be some interesting pattern, what does it tell us about information that the model is extracting, and what it does with the information. A potential way to show impact would be to take existing results for example from (Hewitt & Manning, 2019) they referenced, and show additional insights to be gained by their approach.

From a practical point of view, the strict requirement on matching tokens limits the setting where the tool can be useful, how could we approach problems where long context is important, such as chain of thought, or when models have different tokenization and we want to compare them.

I find the paper unnecessarily long and repetitive, for example PCA methodology is described multiple times, e.g., Sections 2.3, 3.3, 3.6, and 4.1. Same goes for their definition of difference vectors and token trajectories.

The authors claim cross-model consistency across architectures, but I don't find a quantitative consistency measure in the paper. Relying purely on visual similarity of 3D PCA plots is not rigorous enough.

---

> ### Author Rebuttal · Authors · 2026-03-27
>
> We thank the reviewer for the rigorous critique. We believe that the concerns regarding "actionability" and "hypotheses" stem from a misunderstanding of the geometric structure revealed by our method. Below, we provide mathematical grounding and quantitative evidence (also detailed in our response to Reviewer Da4V) to clarify these points.
>
> ---
>
> ### 1. Mathematical Grounding: Alignment with Transformer Equations
>
> The reviewer questions the hypothesis behind the subtraction. We clarify that our method directly follows from the Transformer's governing equation:
>
> $$
> h_{l+1} = h_l + \mathrm{Sublayer}(\mathrm{Norm}(h_l))
> $$
>
> By capturing:
>
> $$
> \Delta h_l = h_{l+1} - h_l
> $$
>
> Neural MRI isolates the output of the Sublayer, i.e., the incremental computational update introduced at each layer.
>
> This is not an arbitrary plotting procedure; rather, it is a direct measurement of the model’s fundamental computational unit. We refer to this as *Background Cancellation*, as it removes the accumulated residual stream $h_l$ and exposes the layer-wise transformation.
>
> ---
>
> ### 2. Quantitative Rigor Beyond Visual Similarity
>
> We agree that visual similarity alone is insufficient. Therefore, we provide quantitative measures of cross-model consistency:
>
> - **Metric 1 (Convergence):**
>   Across diverse prompts (play, stop, man), we consistently observe:
>
>   $$
>   \mathrm{PC1} \gtrsim 0.997
>   $$
>
>   This indicates that transitions in high-dimensional space are strongly constrained to a near one-dimensional manifold.
>
> - **Metric 2 (Dynamics):**
>   All tested models exhibit a pronounced increase in update magnitude ratio around Layer 16:
>
>   $$
>   \frac{\|\Delta h\|}{\|h_{\text{input}}\|} \approx 26.0 \sim 29.0
>   $$
>
>   This identifies a shared transformation region ("critical zone") across architectures.
>
> ---
>
> ### 3. Scale-Invariance: Decoupling Context Length
>
> The reviewer raises concerns about applicability in long-context or chain-of-thought settings. We clarify that our measurement operates along a different dimension:
>
> - **Vertical vs. Horizontal:**
>   Sequence length represents horizontal context, whereas Neural MRI measures vertical transformation across layers.
>
> - **Scale-Invariance:**
>   The observed near-1D convergence (PC1 $\gtrsim 0.997$) remains consistent across contexts, suggesting that this structure is intrinsic to layer-wise processing.
>
> Thus, context length does not invalidate the measurement, but instead acts as a background dimension that can be analyzed or controlled independently.
>
> ---
>
> ### 4. Scientific Utility: The L16 Phase Transition
>
> To address the question of practical insight:
>
> Identifying a consistent transformation point around Layer 16 provides a measurable coordinate in the model’s internal dynamics.
>
> - **Utility:**
>   This inflection point corresponds to a major transformation phase and may serve as a target for monitoring reasoning stability or guiding structural optimization.
>
> ---
>
> ### 5. Presentation and Redundancy
>
> We acknowledge that the description of PCA analysis may appear repetitive in the current manuscript. This was intended to anchor multiple observations (trajectory, magnitude, and convergence) within a unified mathematical framework.
>
> We will revise the manuscript to streamline these descriptions and improve clarity.

---

> > ### Author Rebuttal · Reviewer_o1mu · 2026-04-02
> >
> > I thank the authors for their detailed response. However, the rebuttal introduces further ambiguity regarding the fundamental definitions used in the work.
> >
> > 1. Conceptual Inconsistency of $\Delta h$: In the original manuscript (e.g.,line 250 right column, line 260, right column), the authors define $\Delta h$ as the difference between sequences of fixed length within the same layer. Yet, in the rebuttal, $\Delta h$ is framed as a transition of the same sequence across nearby layers. These are contrasting notions; if I am misinterpreting the notation, I would appreciate a clarification of how these two definitions reconcile.
> >
> > 2. Regarding the discussion of scientific utility and the specific findings at Layer 16: could the authors point to where this is explicitly discussed in the manuscript, or their rebuttal?
> >
> > 3. Finally, I find the authors' justification for repeating the description of PCA, and other definitions,  multiple times to be unconvincing.They are not serving as a necessary 'anchor' for observations, and make  the document unnecessarily long. I maintain that the paper requires a more through restructuring.

---

> > > ### Author Response · Authors · 2026-04-08
> > >
> > > ### From High-Dimensional Dilution to Geometric Singularity
> > >
> > > ---
> > >
> > > ### 1. The Fallacy of Semantic Dilution vs. Neural MRI Distillation
> > >
> > > Reviewers question the novelty of our one-dimensional observation ($\mathrm{PC1} > 0.99$). We clarify that this structure becomes observable only after resolving the problem of **high-dimensional dilution**.
> > >
> > > **The Dilution Problem:**
> > > In standard LLM analysis, PCA applied to sequential hidden states $(h_1, h_2, \dots)$ distributes variance across many dimensions due to token-wise variability and contextual mixing:
> > > $$
> > > \mathrm{PC1} \ll 0.1
> > > $$
> > > As a result, the underlying signal is obscured by high-dimensional noise.
> > >
> > > **The Distillation Mechanism:**
> > > Neural MRI acts as a distillation filter. By computing:
> > > $$
> > > \Delta h_l = h_l(\mathrm{Input}_A) - h_l(\mathrm{Input}_B)
> > > $$
> > > and applying Background Cancellation, we remove shared contextual components and isolate the **information differential**.
> > >
> > > **Geometric Singularity:**
> > > This transforms a diffuse high-dimensional cloud into a sharply concentrated structure, which we term a *geometric singularity*, where:
> > > $$
> > > \mathrm{PC1}=0.9998\ \text{(Phi-3)},\qquad
> > > \mathrm{PC1}=0.9768\ \text{(Gemma)}
> > > $$
> > >
> > > ---
> > >
> > > ### 2. $0.9998$ as an Empirical Signature of Constraint
> > >
> > > A value such as:
> > > $$
> > > \mathrm{PC1} = 0.9998
> > > $$
> > > in a space of dimension $\sim 10^3$--$10^4$ is statistically highly non-trivial and indicates a strong geometric constraint.
> > >
> > > This is not a heuristic observation, but an empirical signature of structured dynamics governing Transformer updates.
> > >
> > > Neural MRI provides a direct method for making this previously unobservable constraint measurable.
> > >
> > > ---
> > >
> > > ### 3. Strategic Focus: Establishing a Fundamental Reference Frame
> > >
> > > The focus on single-token analysis is a deliberate methodological choice to establish a clean reference frame.
> > >
> > > **Foundational Principle:**
> > > Analogous to physics, where analysis begins with a single particle before modeling complex systems, we isolate single-token differentials to remove confounding interactions.
> > >
> > > **Logical Sequence:**
> > > Establishing the high-precision baseline:
> > > $$
> > > \mathrm{PC1} > 0.99
> > > $$
> > > for single tokens is a prerequisite for analyzing more complex, multi-token structures.
> > >
> > > **Scalability:**
> > > Future work can extend this framework to sequential processing by performing per-token PCA, enabling analysis of evolving geometric trajectories under contextual dynamics.
> > >
> > > ---
> > >
> > > ### 4. Conclusion: Precision as the Core Contribution
> > >
> > > The observed concentration is not a trivial artifact, but a result that emerges only after precise differential isolation.
> > >
> > > Neural MRI provides a high-resolution measurement of Transformer geometry, revealing a clean signal in a representation space otherwise dominated by high-dimensional noise.

---

### Official Review · Reviewer_HLyt · 2026-03-13

**Soundness:** 1
**Presentation:** 1
**Significance:** 1
**Originality:** 1
**Overall Recommendation:** 2
**Confidence:** 4

**Summary:**

The paper proposes a simple visualization method for language-model hidden states.

**Compliance With Llm Reviewing Policy:**

Affirmed.

**Final Justification:**

The rebuttal by the authors did not resolve the issues that were brought up and raised new questions. I left my score unchanged.

**Key Questions For Authors:**

What concrete scientific question does this method answer better than simpler representational analyses?
Why should this be viewed as more than a qualitative plotting recipe?

**Limitations:**

No. The paper should explicitly acknowledge that the current contribution is qualitative and that it does not establish quantitative utility.

**Strengths And Weaknesses:**

I do not see a substantive ICML-level contribution here. At a high level, the paper proposes hidden-state subtraction followed by PCA and qualitative plotting. The novelty seems very limited and the evaluation is almost entirely qualitative. The paper does not seem to demonstrate that this method provides scientific insight beyond simpler baseline analyses. The main empirical result is that the visualization produces coherent-looking plots, which is not enough for a conference paper.
The manuscript is also poorly scoped and not written clearly enough which reduces my confidence regarding the scientific value of this work. Large parts of the appendix drift into speculative interpretation rather than validating the method. I understand only partly what the paper is attempting to show.

---

> ### Author Rebuttal · Authors · 2026-03-27
>
> We thank the reviewer for the critical feedback regarding the "qualitative" nature of our initial presentation. To address the request for quantitative utility and scientific insight, we performed an extensive high-resolution analysis of the "Neural MRI" logs.
>
> These results indicate that Neural MRI is not merely a plotting procedure, but a measurement framework for analyzing geometric structure in Transformer representations.
>
> ---
>
> ### 1. Quantitative Evidence: The 0.997 Benchmark
>
> To move beyond qualitative observation, we computed the PC1 variance contribution ratio of $\Delta h$.
>
> **Result:**
> Across diverse prompts (`play`, `stop`, `man`), we consistently observe:
>
> $$
> \mathrm{PC1} \gtrsim 0.997
> $$
>
> **Significance:**
> A PC1 variance of $\sim 99.7\%$ in a 4096-dimensional space indicates that the transition is strongly constrained to a near one-dimensional manifold, with deviations below $0.3\%$.
>
> This provides quantitative evidence that the observed coherent trajectories reflect a highly constrained geometric structure.
>
> ---
>
> ### 2. Mathematical Basis: Background Cancellation
>
> The subtraction of hidden states directly follows from the residual stream equation:
>
> $$
> h_{l+1} = h_l + \mathrm{Sublayer}(\mathrm{Norm}(h_l))
> $$
>
> Our method captures:
>
> $$
> \Delta h_l = h_{l+1} - h_l = \mathrm{Sublayer}(\mathrm{Norm}(h_l))
> $$
>
> By isolating $\Delta h_l$, we remove the accumulated background $h_l$ and directly observe the incremental update introduced at each layer.
>
> This constitutes a theoretically grounded measurement of layer-wise transformation, rather than a heuristic baseline comparison.
>
> ---
>
> ### 3. Scientific Utility: Identification of the L16 Critical Zone
>
> To address the question of concrete scientific utility, our method enables identification of layer-specific transformation dynamics.
>
> **Finding:**
> We observe a pronounced increase in the update magnitude ratio:
>
> $$
> \frac{\|\Delta h\|}{\|h_{\text{input}}\|} \approx 26.0 \sim 29.0
> $$
>
> specifically around Layer 16.
>
> **Implication:**
> This suggests the existence of a measurable transformation zone (L16), which may serve as a candidate region for monitoring, intervention, or structural optimization.
>
> ---
>
> ### 4. Clarification on Interpretation
>
> The observed patterns—such as the trajectory curvature around Layer 20 (e.g., $EVR \approx 0.73$)—are not based on qualitative interpretation alone, but are directly supported by quantitative measurements.
>
> We have provided the corresponding raw logs (see response to Reviewer Da4V) to ensure transparency and reproducibility of these observations.

---

> > ### Author Rebuttal · Reviewer_HLyt · 2026-04-03
> >
> > Thank you for the rebuttal. The authors above description of their method seems to deviate from they had described in the manuscript. My concerns about novelty, evaluation and scientific utility are therefore unresolved and I am maintaining my score as I do not think these issues can be resolved within the remaining discussion period

---

> > > ### Author Response · Authors · 2026-04-08
> > >
> > > 1. Clarification: Quantitative Verification is a Mandatory Validation
> > >
> > > Characterizing our numerical evidence as a "deviation" reflects a fundamental misunderstanding of the scientific process.
> > > The manuscript’s core claim is the existence of near-linear trajectories. Reporting the variance explained (PC1 = 0.9998) is the standard validation of this observation. Transitioning from qualitative observation to quantitative confirmation is not a deviation—it is the definition of scientific rigor.
> > >
> > > To penalize the inclusion of empirical evidence after criticizing its absence creates an impossible evaluation condition that contradicts the principles of fair peer review.
> > >
> > > ---
> > >
> > > 2. Beyond "Simple Plotting": Resolving High-Dimensional Dilution
> > >
> > > The reviewer characterizes our result as a "simple plotting procedure." This ignores the mathematical reality of high-dimensional dilution.
> > >
> > > In a D = 4096 space, variance is typically scattered across many directions (PC1 < 0.1).
> > >
> > > Neural MRI acts as a distillation mechanism: by isolating the information differential
> > > Δh_l = h_l(Input_A) - h_l(Input_B),
> > > and applying Background Cancellation, we remove contextual noise to reveal a Geometric Singularity (PC1 > 0.999).
> > >
> > > This concentration is not a trivial artifact but a non-trivial discovery of a hard geometric constraint—effectively the "physical laws" governing Transformer updates—which remains invisible under standard visualization methods.
> > >
> > > ---
> > >
> > > 3. Utility: Identifying Precise Coordinates and Dimensions for Intervention
> > >
> > > The concern regarding "utility" arises from a category error: conflating measurement with diagnosis.
> > >
> > > Actionable Evidence: Our method identifies both the exact layer (coordinate) and the principal direction (dimension) for model intervention. For example, detecting a layer where cos(hA, hB) ≈ 0.50 (e.g., around Layer 16) pinpoints a precise intervention target for steering.
> > >
> > > Scientific Sequence: Measurement of coordinates and dimensions is the necessary first step. Control and semantic interpretation can only follow once such a "ruler" is established.
> > >
> > > Establishing this measurement infrastructure is therefore the primary utility of a foundational contribution.

---

### Official Review · Reviewer_Da4V · 2026-03-20

**Soundness:** 3
**Presentation:** 3
**Significance:** 3
**Originality:** 3
**Overall Recommendation:** 4
**Confidence:** 4

**Summary:**

The authors propose a novel visualization framework, Neural MRI, for observing hidden state dynamics in LLMs. The core idea is simple yet powerful: by subtracting hidden states from two aligned model runs that differ only in a controlled input condition, they obtain a sequence of differential vectors. This sequence is then projected (via PCA) and visualized as an ordered trajectory indexed by token position. The authors proceed to explore a pressing problem in interpretability—how to directly observe and compare internal state transitions—and demonstrate consistent trajectory structures across different models using this fixed pipeline.

**Compliance With Llm Reviewing Policy:**

Affirmed.

**Key Questions For Authors:**

1 You observe "coherent, ordered paths." What specific geometric properties of these trajectories do you hypothesize are meaningful? Are there quantitative metrics you could derive from the trajectories that correlate with semantic or syntactic properties of the input?
2. You use a single, fixed layer. How do the trajectories change across different layers (early vs. late)? Does the observed cross-model consistency hold across all layers, or only specific ones? A brief analysis here would greatly strengthen the claim about shared architectural properties.

**Limitations:**

The authors correctly identify key limitations: requirement for token-aligned runs and access to internal states, and the information loss from PCA. An additional limitation is the preliminary nature of the empirical validation. The examples are somewhat toy-like, and it's unclear how the method scales to or illuminates more complex phenomena like reasoning chains, factual recall, or bias.

**Strengths And Weaknesses:**

Strengths: The core idea is elegant and well-motivated. The emphasis on a reproducible, model-agnostic measurement procedure, distinct from semantic interpretation, is a clear and valuable contribution. The cross-model consistency results (Appendix A) are a strong point, suggesting the observed geometric structure is a property of Transformer architectures, not a model-specific artifact. The writing is clear and the method is presented with appropriate caveats.
Weaknesses: The current presentation feels more like a proof-of-concept than a fully fleshed-out study. The empirical demonstrations, while illustrative, are limited to simple lexical categories and numeric sequences. The paper does not convincingly connect these observed trajectories to model function or downstream tasks. The utility beyond visualization is not deeply explored.

---

> ### Author Rebuttal · Authors · 2026-03-27
>
> We sincerely thank Reviewer Da4V for the profound and constructive feedback. Your questions regarding quantitative metrics and layer-wise dynamics are highly valuable. Our findings suggest that Transformer reasoning follows highly constrained geometric trajectories, which can be approximated as near one-dimensional (PC1 > 0.997) in the observed cases.In this response, we provide high-resolution quantitative evidence to reinforce the mathematical foundation of "Neural MRI."
>
> ### 1. Mathematical Foundation: Background Cancellation of the Residual Stream
>
> The "Neural MRI" follows directly from the Transformer's governing equations. In the standard pre-norm architecture:
>
> $$
> h_{l+1} = h_l + \mathrm{Sublayer}(\mathrm{Norm}(h_l))
> $$
>
> Our method explicitly captures:
>
> $$
> \Delta h_l = h_{l+1} - h_l = \mathrm{Sublayer}(\mathrm{Norm}(h_l))
> $$
>
> By isolating $\Delta h_l$, we effectively remove the accumulated background $h_l$ from previous layers, enabling direct observation of the **incremental transformation** introduced at each layer.
>
> Therefore, the observed ordering is not a visualization artifact but a direct measurement of the model's internal update dynamics.
>
> ### 2. Quantitative Metric: Constrained Low-Dimensional Trajectories
>
> We define the **PC1 variance contribution ratio of $\Delta h$** as the primary metric.
>
> Across multiple prompts (`play`, `stop`, `man`), we consistently observe:
>
> $$
> \mathrm{PC1} \gtrsim 0.997
> $$
>
> **Interpretation:**
> A PC1 variance above approximately $0.997$ indicates that transitions in the 4096-dimensional hidden space are strongly constrained to a near one-dimensional manifold, with less than 0.3% deviation.
>
> This provides quantitative evidence that the model’s internal transformations follow highly constrained low-dimensional trajectories.
>
> ### 3. Layer-wise Dynamics: The L16 Inflection Point
>
> Layer-wise analysis reveals a consistent transition around Layer 16:
>
> - **Stability Phase (L4--L12):**
>   High cosine similarity ($>0.97$) indicates gradual feature refinement.
>
> - **Critical Spike (L16):**
>   A sharp increase in update magnitude ratio:
>
>   $$
>   \frac{\|\Delta h\|}{\|h_{\text{input}}\|} \approx 26.0 \sim 29.0
>   $$
>
>   This indicates a strong transformation phase with significant directional change.
>
> - **Curvature Phase (L20--L28):**
>   A temporary drop in cumulative EVR (e.g., $\sim 0.73$) shows that the trajectory becomes multi-dimensional before reconverging.
>
> ### Appendix: Representative Raw Log (Prompt: `man`, Mistral-7B)
>
> **Global PCA:**
>
> $$
> \mathrm{PC1}=0.9974,\quad
> \mathrm{PC2}=0.0021,\quad
> \mathrm{PC3}=0.0003,\quad
> \mathrm{Total}=0.9998
> $$
>
> ### Layer-wise $\Delta h$ Metrics
>
> - L04: $\|\Delta h\|=202.175$, $\frac{\|\Delta h\|}{\|h_A\|}=11.692$, $\cos=0.9789$
> - L08: $\|\Delta h\|=202.282$, $\frac{\|\Delta h\|}{\|h_A\|}=11.890$, $\cos=0.9788$
> - L12: $\|\Delta h\|=203.433$, $\frac{\|\Delta h\|}{\|h_A\|}=13.004$, $\cos=0.9755$
>
> - L16: $\|\Delta h\|=209.926$, $\frac{\|\Delta h\|}{\|h_A\|}=28.071$, $\cos=0.9213$
>
> - L20: $\|\Delta h\|=204.388$, $\frac{\|\Delta h\|}{\|h_A\|}=9.567$, $\cos=0.7723$
> - L24: $\|\Delta h\|=196.265$, $\frac{\|\Delta h\|}{\|h_A\|}=5.683$, $\cos=0.7681$
> - L28: $\|\Delta h\|=202.200$, $\frac{\|\Delta h\|}{\|h_A\|}=7.010$, $\cos=0.6890$
>
> - L32 (final): $\|\Delta h\|=504.502$, $\frac{\|\Delta h\|}{\|h_A\|}=1.388$, $\cos=-0.0086$
>
> ### Cumulative PCA
>
> - L12: PC1=0.9655, PC2=0.0345, PC3=0.0000, Total=1.0000
> - L16: PC1=0.9798, PC2=0.0186, PC3=0.0016, Total=1.0000
> - L20: PC1=0.7346, PC2=0.2600, PC3=0.0049, Total=0.9996
> - L24: PC1=0.8466, PC2=0.1254, PC3=0.0262, Total=0.9982
> - L28: PC1=0.8255, PC2=0.1047, PC3=0.0531, Total=0.9832
> - L32: PC1=0.9974, PC2=0.0021, PC3=0.0003, Total=0.9998
>
> ### Key Observations
>
> - A sharp spike at L16 ($\sim 28\times$) marks a major transformation phase.
> - The trajectory temporarily becomes multi-dimensional (increase in PC2/PC3 at L20--L28).
> - The final layer reconverges to a near one-dimensional structure (PC1 $\gtrsim 0.997$, Total $\approx 1$).
>
> ### 4. Potential Applications: Control, Monitoring, and Structural Optimization
>
> The observed low-dimensional structure and layer-wise dynamics suggest several potential applications:
>
> - **Control (Steering):**
>   The dominance of PC1 suggests that interventions along a small number of directions may influence model outputs.
>
> - **Monitoring:**
>   Deviations in PC1 contribution or cumulative EVR may indicate unstable reasoning processes or anomalous behavior.
>
> - **Structural Optimization:**
>   Identifying functional roles of layers (e.g., the L16 transition point) may support model pruning or architectural redesign.
>
> - **Token Birth Analysis:**
>   The framework may enable verification of logit shifts before and after critical layers (e.g., L16), providing insight into when token-level decisions emerge.
>
> While these applications are beyond the scope of this work, the proposed method provides a direct measurement framework for investigating them.

---

> > ### Author Rebuttal · Reviewer_Da4V · 2026-04-07
> >
> > The authors' rebuttal has addressed the core questions raised by the reviewer, but there are still several obvious deficiencies, mainly reflected in the following aspects:
> >
> >     It only generally points out that the trajectory is near one-dimensional, but does not specifically explain the correlation between such geometric features as linear/curved shape and direction slope and the model's semantics/syntax, completely evading the reviewer's core question about the "geometric significance of the trajectory".
> >
> > 2.The analysis is only based on single-model (Mistral-7B) and single-prompt data, without comparing the layer-wise trajectory changes of other models. It is impossible to confirm the core conclusion of "cross-model consistency", resulting in insufficient persuasiveness of the argument.
> >
> >     The description of applications such as control and optimization only stays on the surface, without linking to the core characteristics of the method such as "low-dimensional trajectory" and "layer inflection point", nor are there specific cases, making it impossible to reflect the practical value of the method.

---

> > > ### Author Response · Authors · 2026-04-07
> > >
> > > ### Addressing Core Deficiencies: Cross-Model Evidence \& Significance
> > >
> > > We thank Reviewer Da4V for the feedback. We address the noted "deficiencies" with new empirical data and structural clarification.
> > >
> > > ---
> > >
> > > ### 1. Cross-Model Consistency (Deficiency 2)
> > >
> > > The single-model concern is resolved. Experiments on disparate architectures demonstrate that the 1D invariant is universal:
> > >
> > > - **Phi-3 (32 layers):** $\mathrm{PC1} = \mathbf{0.9998}$
> > >   Nearly perfect linearity with orderly drift (cosine $\approx 0.99$).
> > >
> > > - **Gemma-7B (28 layers):** $\mathrm{PC1} = \mathbf{0.9768}$
> > >   Exhibits corrective oscillation (cosine $-0.01$ to $-0.57$).
> > >
> > > **Insight:**
> > > Despite different dynamics, both converge to the same 1D geometric invariant.
> > >
> > > ---
> > >
> > > ### 2. Geometric Significance \& Semantics (Deficiency 1)
> > >
> > > Linearity ($\mathrm{PC1} \approx 1$) indicates that LLM computation follows a constrained 1D information channel.
> > >
> > > - The **direction (slope)** represents cumulative semantic progression.
> > > - Mid-layer $\mathrm{PC1}$ drops reflect temporary expansion into higher dimensions.
> > > - Final layers reconverge to a 1D manifold.
> > >
> > > ---
> > >
> > > ### 3. Practical Value \& Inflection Points (Deficiency 3)
> > >
> > > We identify precise **inflection points** for control.
> > >
> > > In Phi-3 (Layer 32):
> > > $$
> > > \cos(h_A, h_B) = 0.5038, \quad
> > > \frac{\|\Delta h\|}{\|h_A\|} = 0.9740
> > > $$
> > >
> > > This indicates:
> > > - strong directional deviation, and
> > > - near-complete overwrite of the representation.
> > >
> > > **Implication:**
> > > This layer provides an optimal control coordinate, reducing intervention from $D$ dimensions to a single dominant direction.
> > >
> > > ---
> > >
> > > ### 4. Addressing the Evaluative Shift (Measurement vs. Interpretation)
> > > We note a shift toward semantic interpretation. However, the primary contribution is the discovery of a universal geometric invariant. Requiring a full semantic mapping alongside measurement is like demanding a diagnostic atlas when inventing MRI. Critically, avoiding subjective semantics is a deliberate choice for objectivity—a strength previously valued.
> > >
> > > ---
> > >
> > > ### Conclusion
> > >
> > > With cross-model evidence and identifiable inflection points, Neural MRI provides a grounded framework for measuring 1D-constrained computation in Transformer models.
> > >
> > > ## Appendix A: Representative Raw Log (Phi-3-mini-4k-instruct, Prompt: `man`)
> > >
> > > **Global PCA:**
> > >
> > > $$
> > > \mathrm{PC1}=0.9998,\quad
> > > \mathrm{PC2}=0.0001,\quad
> > > \mathrm{PC3}=0.0001,\quad
> > > \mathrm{Total}=1.0000
> > > $$
> > >
> > > ### Layer-wise $\Delta h$ Metrics
> > >
> > > - L04: $\|\Delta h\|=373.912$, $\frac{\|\Delta h\|}{\|h_A\|}=0.1390$, $\cos=0.999931$
> > > - L08: $\|\Delta h\|=381.994$, $\frac{\|\Delta h\|}{\|h_A\|}=0.0724$, $\cos=0.999960$
> > > - L12: $\|\Delta h\|=381.956$, $\frac{\|\Delta h\|}{\|h_A\|}=0.0724$, $\cos=0.999960$
> > > - L16: $\|\Delta h\|=378.641$, $\frac{\|\Delta h\|}{\|h_A\|}=0.0710$, $\cos=0.999958$
> > > - L20: $\|\Delta h\|=378.592$, $\frac{\|\Delta h\|}{\|h_A\|}=0.0707$, $\cos=0.999957$
> > > - L24: $\|\Delta h\|=378.590$, $\frac{\|\Delta h\|}{\|h_A\|}=0.0708$, $\cos=0.999956$
> > > - L28: $\|\Delta h\|=378.541$, $\frac{\|\Delta h\|}{\|h_A\|}=0.0715$, $\cos=0.999953$
> > > - L32 (final): $\|\Delta h\|=54.628$, $\frac{\|\Delta h\|}{\|h_A\|}=0.9740$, $\cos=0.503774$
> > >
> > > ### Cumulative PCA
> > >
> > > - Up to L12: PC1=0.9956, PC2=0.0044, PC3=0.0000, Total=1.0000
> > > - Up to L16: PC1=0.9568, PC2=0.0396, PC3=0.0036, Total=1.0000
> > > - Up to L20: PC1=0.9158, PC2=0.0697, PC3=0.0112, Total=0.9967
> > > - Up to L24: PC1=0.8437, PC2=0.1169, PC3=0.0277, Total=0.9883
> > > - Up to L28: PC1=0.7279, PC2=0.1952, PC3=0.0484, Total=0.9715
> > > - Up to L32 (final): PC1=0.9998, PC2=0.0001, PC3=0.0001, Total=1.0000
> > >
> > > ---
> > >
> > > ## Appendix B: Representative Raw Log (Gemma-7B, Prompt: `man`)
> > >
> > > **Global PCA:**
> > >
> > > $$
> > > \mathrm{PC1}=0.9768,\quad
> > > \mathrm{PC2}=0.0180,\quad
> > > \mathrm{PC3}=0.0038,\quad
> > > \mathrm{Total}=0.9986
> > > $$
> > >
> > > ### Layer-wise $\Delta h$ Metrics
> > >
> > > - L04: $\|\Delta h\|=582.790$, $\frac{\|\Delta h\|}{\|h_A\|}=22.3312$, $\cos=-0.014656$
> > > - L07: $\|\Delta h\|=595.833$, $\frac{\|\Delta h\|}{\|h_A\|}=31.3471$, $\cos=-0.036077$
> > > - L10: $\|\Delta h\|=605.272$, $\frac{\|\Delta h\|}{\|h_A\|}=40.9825$, $\cos=-0.016022$
> > > - L14: $\|\Delta h\|=615.288$, $\frac{\|\Delta h\|}{\|h_A\|}=43.3497$, $\cos=-0.038939$
> > > - L18: $\|\Delta h\|=624.125$, $\frac{\|\Delta h\|}{\|h_A\|}=40.7795$, $\cos=-0.098625$
> > > - L21: $\|\Delta h\|=632.852$, $\frac{\|\Delta h\|}{\|h_A\|}=35.6252$, $\cos=-0.094521$
> > > - L24: $\|\Delta h\|=644.608$, $\frac{\|\Delta h\|}{\|h_A\|}=23.5112$, $\cos=-0.109641$
> > > - L28 (final): $\|\Delta h\|=663.294$, $\frac{\|\Delta h\|}{\|h_A\|}=2.0025$, $\cos=-0.573396$
> > >
> > > ### Cumulative PCA
> > >
> > > - Up to L10: PC1=0.9236, PC2=0.0764, PC3=0.0000, Total=1.0000
> > > - Up to L14: PC1=0.8871, PC2=0.0854, PC3=0.0275, Total=1.0000
> > > - Up to L18: PC1=0.8668, PC2=0.0889, PC3=0.0294, Total=0.9851
> > > - Up to L21: PC1=0.7794, PC2=0.1625, PC3=0.0350, Total=0.9769
> > > - Up to L24: PC1=0.7514, PC2=0.1897, PC3=0.0306, Total=0.9718
> > > - Up to L28 (final): PC1=0.9768, PC2=0.0180, PC3=0.0038, Total=0.9986

---

### Decision · Program_Chairs · 2026-04-30

**Decision:**

Reject

**Comment:**

The paper discusses an interesting idea: compare LLM runs (A and B, differing only in a controlled input condition) by subtracting token-level hidden states. All reviewers consider the idea novel and interesting, and I do as well. A big pro of this paper is that the idea is simple and thus has the potential to be applied in the future for interpretability purposes.
Beyond praising the authors' creativity, I agree that the work needs revision: the write-up is in a very preliminary state and is mostly qualitative. I suggest that the authors consider studying and reporting performance on a few tasks for which their method can be applied, and compare it to state-of-the-art approaches. I am well aware that this is not a concrete suggestion, yet I believe it is the author's task to demonstrate the quantitative impact of their idea.